



# Continental heat storage: Contributions from ground, inland waters, and permafrost thawing

Francisco José Cuesta-Valero[1,2], Hugo Beltrami[3,4], Almudena García-García[1,2], Gerhard Krinner[5], Moritz Langer[6,7], Andrew H. MacDougall[8], Jan Nitzbon[6,9], Jian Peng[1,2], Karina von Schuckmann[10], Sonia I. Seneviratne[11], Noah Smith[12], Wim Thiery[13], Inne Vanderkelen[13], Tonghua Wu[14]

[1]Department of Remote Sensing, Helmholtz Centre for Environmental Research, Leipzig, 04318, Germany.
[2]Remote Sensing Centre for Earth System Research, Leipzig University, 04103, Leipzig, Germany.
[3]Climate & Atmospheric Sciences Institute and Department of Earth Sciences, St. Francis Xavier University, Antigonish, B2G 2W5, Canada.
[4]Département des sciences de la Terre et de l'atmosphère, Université du Québec à Montréal, Montréal, Québec, Canada.
[5]CNRS senior scientist (Directeur de Recherche), LGGE Grenoble, France.
[6]Permafrost Research Section, Alfred Wegener Institute Helmholtz Centre for Polar and Marine Research, Potsdam, Germany.
[7]Geography Department, Humboldt-Universität zu Berlin, Berlin, Germany.
[8]Climate & Environment Program, St. Francis Xavier University Antigonish, Nova Scotia, Canada B2G 2W5.
[9]Paleoclimate Dynamics Section, Alfred Wegener Institute Helmholtz Centre for Polar and Marine Research, Bremerhaven, Germany.
[10]Mercator Ocean International, Toulouse, 31400, France.
[11]Institute for Atmospheric and Climate Science, ETH Zurich, Zurich, 8092, Switzerland.
[12]Department of Mathematics, University of Exeter, Exeter, United Kingdom.
[13]Department of Hydrology and Hydraulic Engineering, Vrije Universiteit Brussel, Brussels, 1050, Belgium.
[14]Cryosphere Research Station on the Qinghai–Tibet Plateau, State Key Laboratory of Cryospheric Science, Northwest Institute of Eco–Environment and Resources (NIEER), Chinese Academy of Sciences (CAS), Lanzhou, 730000, China.

*Correspondence to*: Francisco José Cuesta-Valero (francisco-jose.cuesta-valero@ufz.de)

**Abstract.** Heat storage within the Earth system is a fundamental metric to understand climate change. The current energy imbalance at the top of the atmosphere causes changes in energy storage within the ocean, the atmosphere, the cryosphere, and the continental landmasses. After the ocean, heat storage in land is the second largest term of the Earth heat inventory, affecting physical processes relevant to society and ecosystems, such as the stability of the soil carbon pool. Here, we present an update of the continental heat storage combining for the first time the heat in the land subsurface, inland water bodies, and permafrost thawing. The continental landmasses stored $23.9\pm0.4\times10^{21}$ J during the period 1960-2020, but the distribution of heat among the three components is not homogeneous. The ground stores ~90 % of the continental heat storage, with inland water bodies and permafrost degradation accounting for ~0.7 % and ~9 % of the continental heat, respectively. Although the inland water bodies and permafrost soils store less heat than the ground, we argue that their associated climate phenomena justify their monitoring and inclusion in the Earth heat inventory.



## 1 Introduction

Anthropogenic changes in atmospheric composition have contributed to sustain the positive radiative imbalance measured at the top of the atmosphere, leading to an accumulation of heat within the Earth system (Levitus et al., 2005; Church et al., 2011; Hansen et al., 2011; von Schuckmann et al., 2020; Forster et al., 2021). The ocean, atmosphere, cryosphere, and continental landmasses have shown a marked increase in heat storage since the 1960s, with the ocean accounting for about 89% of the total heat storage, the continental subsurface for 6%, the cryosphere for 4% and the atmosphere for 1% (von Schuckmann et al., 2020). Continental heat storage has consistently ranked as the second largest term of the Earth heat inventory only after the ocean, showing similar values to the heat uptake by the cryosphere (Levitus et al., 2005; Church et al., 2011; Hansen et al., 2011; von Schuckmann et al., 2020). These previous analyses included estimates of heat storage within the global subsurface retrieved from inversions of temperature-depth profiles measured around the world (Beltrami et al., 2002; Cuesta-Valero et al., 2021a). Subsurface temperature profiles record long-term changes in the surface energy balance as perturbations of subsurface temperatures (Beltrami, 2002). If heat diffusion through the ground occurs in a conductive regime, the original changes in ground heat flux at the surface can be retrieved by inverting the measured temperature profiles (Beltrami, 2001; Beltrami et al., 2002; Cuesta-Valero 2021a), from which the ground heat storage can be estimated. Nevertheless, these inversions of subsurface temperature profiles only capture changes in the subsurface thermal regime due to conductive heat diffusion, and other processes should be considered in order to estimate the total continental heat storage.

Phase change in permafrost soils involves the high latent heat of fusion of ice and it is not captured in inversions of subsurface temperature profiles, thus the heat used to thaw ground ice could be a relevant contributor to continental heat storage, at least at higher latitudes. Permafrost soils underline 11 % of the global exposed land surface (Obu, 2021), with continuous permafrost warming by 0.4 °C, and permafrost in the discontinuous zone warming by 0.2 °C between 2007 and 2016 (Biskaborn et al., 2019; Fox-Kemper et al., 2021). Active layer thickness is also increasing at most measurement locations around the world (Smith et al., 2022b). Additionally, global climate simulations project a decrease in global permafrost extension from 10 % to more than 80 % due to thawing by the end of the 21st century, depending on future greenhouse gas emissions (Koven et al., 2013; Slater & Lawrence, 2013; Burke et al., 2020; Hermoso de Mendoza et al.; 2020; Steinert et al., 2021). Consequently, permafrost heat uptake is expected to increase in the following decades.

Similarly, a large amount of energy is required to warm lakes, rivers, and artificial reservoirs due to the high heat capacity of water, which may constitute another relevant contribution to continental heat storage not included in previous analyses. Inland surface water bodies extend through a considerable part of the land surface. For instance, natural lakes cover ~2 % (2,662,040 km$^2$) of the global continental surface (Messager et al., 2016; Vanderkelen et al., 2020). Rivers and lakes have warmed by 1 °C dec$^{-1}$ and 0.45 °C dec$^{-1}$, respectively, in recent times, resulting in an a reduction of ~25 % in ice cover, and there is high confidence that these trends are going to maintain by the 21st century according to the IPCC Sixth Assessment Report (Douville et al., 2021). Furthermore, previous estimates of heat flux in global inland surface water bodies



from a multimodel ensemble of simulations yielded ~121 mW m$^{-2}$ (1 mW m$^{-2}$ = 0.001 W m$^{-2}$) for the period 1991-2020
(Vanderkelen et al., 2020), which is similar to the ground heat flux determined from subsurface temperature profiles.

        Here, we quantify the continental heat storage by combining ground heat storage, heat uptake by inland water
bodies, and heat used for thawing permafrost. Heat storage from these three components is estimated from 1960 to 2020 and
compared to previous estimates of the Earth heat inventory. These estimates of continental heat storage will contribute to
updating the global Earth heat inventory defined in von Schuckmann et al. (2020). We also argue about the importance of
monitoring all three components of the continental heat storage in the future due to the implications of changes in heat
content within these subsystems for society and ecosystems.

## 2 Data and Methods

### 2.1 Estimates of Ground Heat Storage

Ground heat fluxes are estimated from subsurface temperature profiles, consisting of measurements of temperature with
depth usually performed in holes that were drilled for mining prospecting campaigns, and thus unevenly distributed across
the global land surface. These profiles are typically described by two components: a quasi-equilibrium temperature profile,
and the propagation of recent variations in the surface energy balance (Beltrami, 2002). The quasi-equilibrium profile
corresponds to the temperature profile in an equilibrium state, that is, with a constant surface temperature and geothermal
gradient. Heat flow from the Earth interior is constant at temporal scales of millions of years, thus the local geothermal
gradient can be considered as constant (Jaupard & Mareschal, 2010). However, recent changes in total radiation reaching the
land surface (Wild et al., 2015) ensure that local surface temperatures are not constant in the long-term. A common approach
to estimate the quasi-equilibrium profile consists in performing a linear regression analysis of the deepest part of each
profile, as this is the part least affected by recent changes in surface conditions (Cuesta-Valero et al., 2019). Thereby, the
geothermal gradient is assumed to correspond to the slope of this regression analysis, while the extrapolation of the fitted
line to the surface is considered as the long-term past surface temperature. Variations of the surface energy balance are
assumed to propagate into the ground following the one-dimensional heat diffusion equation, and are recorded in the profile
as alterations of the quasi-equilibrium profile (Carslaw & Jaeger, 1959). Therefore, this signature of changes in surface
conditions on subsurface temperatures can be estimated by subtracting the quasi-equilibrium profile to the measured log,
obtaining an anomaly profile. Ground heat flux histories retrieved in this analysis are based on inverting the anomaly profile
of each individual subsurface temperature profile.

        We invert subsurface temperature profiles from the Xibalbá dataset (Cuesta-Valero et al., 2021a,b) to estimate the
global long-term ground heat flux history. The Xibalbá dataset consists of 1079 subsurface temperature profiles measured
around the world, with a larger number of profiles in the mid latitudes of the northern hemisphere. Xibalbá logs have been
harmonised to include temperature records from 15 m to 300 m of depth. Ensuring that all logs are truncated at the same





depth is crucial to obtain temperature and heat flux estimates relative to the same temporal reference, approximately the

period 1300-1700 in this analysis. This period of reference arises from the depth range used to perform the linear regression

analysis from which the corresponding quasi-equilibrium temperature profile from each individual log is determined in this

case the depths between 200 m and 300 m (see Cuesta-Valero et al., 2019 for details about the relationship between depth

range and period of reference). Once the quasi-equilibrium profile is estimated, the corresponding anomaly profile is

retrieved as explained above.

Ground surface temperature histories are estimated from individual Xibalbá profiles using a Singular Value

Decomposition (SVD) algorithm (Lanczos, 1961) to invert each anomaly profile. Inversions performed by this SVD

algorithm are common in the literature (Beltrami & Mareschal, 1992; Mareschal & Beltrami, 1992; Clauser & Mareschal,

1995; Beltrami et al., 2015; Jaume-Santero et al., 2016; Pickler et al., 2016), and have shown robust results in experiments

designed to test their ability to retrieve past changes in global surface temperature (González-Rouco et al., 2006; González-

Rouco et al., 2009; García-García et al., 2016; Melo-Aguilar et al., 2018). Ground heat flux histories are then retrieved from

each ground surface temperature history using the technique developed in Wang & Brass (1999) from a half-order derivative

approach:

$$G\left(t_N\right) = \frac{2\lambda}{\sqrt{\pi\alpha}} \sum_k^{N-1} \frac{T_{k+1} - T_k}{t_{k+1} - t_k} \cdot \left[\sqrt{t_N - t_k} - \sqrt{t_N - t_{k+1}}\right], \tag{1}$$

with $\alpha$ the thermal diffusivity of the medium, $\lambda$ the thermal conductivity, $G\left(t_N\right)$ the ground heat flux at the time $t_N$, $T_k$ the

ground surface temperature history at the $k$-th time step, and $t_k$ the time at the $k$-th time step. This approach has been used

extensively in the literature to derive global ground heat flux histories, and has shown good results against other techniques

(Beltrami, 2001; Beltrami et al., 2002; Bennett et al., 2008; Cuesta-Valero et al., 2021a). Here, we consider thermal

diffusivities ranging from $0.5\times10^{-6}$ m$^2$ s$^{-1}$ to $1.5\times10^{-6}$ m$^2$ s$^{-1}$, and thermal conductivities between 2.5 W m$^{-1}$ K$^{-1}$ and 3.5 W m$^{-1}$

K$^{-1}$ to perform the inversions and to estimate ground heat flux histories, which are typical values in the literature.

These SVD inversions are combined with a bootstrap sampling strategy to retrieve the 2.5[th], 50[th], and 97.5[th]

percentiles of the spatially aggregated heat flux histories (Efron, 1987; DiCiccio & Efron, 1996; Davison & Hinkley, 1997).

The bootstrap method consists in estimating global mean ground heat fluxes from populations of 1079 elements (i.e., the

number of profiles considered), with each element a ground heat flux history from a Xibalbá profile using randomly selected

values for thermal diffusivity and thermal conductivity (see ranges above), as well as a random quasi-equilibrium

temperature profile. This random quasi-equilibrium profile is chosen from the Gaussian distribution of long-term mean

surface temperature and geothermal gradient retrieved from the linear regression analysis performed in the deepest 100 m of

the corresponding profile, as explained above. This process is repeated 1000 times to obtain an ensemble of global heat flux

averages, considering the 50[th] percentile of the ensemble as the best estimate of global ground heat flux, and the 2.5[th] and





97.5<sup>th</sup> percentiles as the 95 % confidence interval. A detailed description of the bootstrapping sampling approach combined with the SVD algorithm can be found in Cuesta-Valero et al. (2022).

Ground heat storage is estimated as the accumulated heat flux since 1960, considering the global land area without Greenland and Antarctica as there are no measured profiles in these areas. Since the number of measured profiles decreases sharply after 2000, we extrapolate the trend of ground heat flux for the period 1970-2000 to fill the period 2000-2020 with data, as in von Schuckmann et al. (2020).

**2.2 Estimates of Permafrost Heat Storage**

Latent heat stored in permafrost due to melting of ground ice is evaluated based on extensive parameter ensemble simulations using the CryoGridLite permafrost model (Langer et al., 2022; Nitzbon et al., 2022). The model uses an implicit, iterative, backward Euler scheme to solve the heat transfer equation with phase change in mixed enthalpy form (Swaminathan & Voller, 1992). Daily average enthalpy and liquid water content profiles are calculated to a depth of 550 m

with a spatial resolution of 1° (per grid cell) for the Arctic permafrost region. Thermal offsets at the ground surface caused by snow are represented by a dedicated snow scheme accounting for regional snow characteristics (Sturm et al., 2010). Ground stratigraphies which determine both the thermal properties of the ground as well as the amount and location of ground ice are derived based on soil stratigraphy parameterizations developed for the SURFEX land surface model (Le Moigne et al., 2009). Required input data are extracted from multiple global datasets such as soil sand and clay fractions

(Masson et al., 2003; Faroux et al., 2013), the soil organic carbon content (Hugelius et al., 2013), and the soil thickness (Pelletier et al., 2016).

Uncertainties in soil stratigraphies affecting latent heat storage are primarily determined by the amount and distribution of ground ice. Such uncertainties are accounted for by ensemble parameter simulations (N=100) using a Monte Carlo approach that randomly varies soil thickness as well as the thickness of soil layers with different ice saturation (Langer

et al., 2022; Nitzbon et al., 2022). Additional data on excess soil ice are included by increasing the soil ice content for 50% of the ensemble members by a random fraction based on the soil ice categories given in the map by Brown et al. (2022). The climate forcing of the simulations performed is based on daily mean surface temperatures and daily snowfall amounts. The climate forcing used is a synthetic time series combining Commonwealth Scientific and Industrial Research Organisation (CSIRO) paleoclimate simulations (500 CE - 1979) based on the Mk3L climate system model (Phipps et al., 2013) with

reanalysis data (1979 - 2019) based on ERA-Interim (Dee et al., 2011). Both climate time series are harmonised using an anomaly approach to extend the last decade of the reanalysis data into the past. The lower boundary condition at 550 m depth is set to a local geothermal heat flux according to the Global Map of Solid Earth Surface Heat Flow (Davies, 2013). Here, the period from 500 CE to 1960 is considered sufficient to bring the model to dynamic equilibrium after initialization with steady-state conditions (averaged for 500 to 600 CE).

The results of the simulations are analysed by integrating the daily liquid soil water content profiles with depth to obtain the total average annual liquid water content per square metre. Multiplying this water content by the volumetric latent





heat of fusion of water ($334×10^6$ kJ m$^{-3}$) yields the average annual latent heat uptake per square metre which is multiplied with the land area (excluding the surface water area) contained within each model grid cell. The uncertainties caused by uncertain ground ice distributions are provided as average standard deviation calculated from the ensemble.

**2.3 Estimates of Inland Water Heat Storage**

The heat storage by inland waters, including natural lakes, reservoirs and rivers, is estimated for the period 1900-2021 by combining water temperature anomalies with volume estimates. To this end, we use a combination of global-scale lake models, global hydrological models and Earth System Model (ESM) simulations from the Inter-Sectoral Impact Model Intercomparison Project phase 2b (ISIMIP2b, Frieler et al., 2017; Golub et al., 2022), with the methods described in Vanderkelen et al. (2020). To derive lake temperature profiles, we expand the global lake model ensemble consisting of three global lake models (CLM45, ALBM and Simstrat-UoG), each providing simulations driven by bias-adjusted atmospheric forcing from four ESMs (GFDL-ESM2M, HadGEM2-ES, IPSL-CM5A-LR and MIROC5) with four simulations using an additional global lake model, the General Ocean Turbulence Model (GOTM; Sachse et al., 2014) driven by the same ESMs. In total, the ensemble contains 16 global lake simulations, providing lake temperature profiles (Table 3) for the period 1900 to 2021 on a 0.5° by 0.5° grid. These simulations are combined with global gridded lake depths from the Global Lake Database v.3 (GLDB; Choulga et al., 2019) rasterized global lake and reservoir area from HydroLAKES and GRanD databases (Messager et al., 2016; Lehner et al., 2011), as described in Vanderkelen et al. (2020).

Different from the cylindrical lake assumption of Vanderkelen et al. (2020), in which the grid cell lake volume is calculated by multiplying lake area and depth, we determine lake and reservoir volumes by estimating lake morphometry with the volume development parameter ($V_d$), which is a well-established geometric approach (Johansson et al., 2007; Håkanson & Hakanson, 1977). The $V_d$ parameter represents the extent to which the lake volume deviates from the volume of a cylinder, thereby indicating whether the lake morphometry is concave or convex. We employ a global constant $V_d$ value of 1.19, which is the median $V_d$ derived from the 1 427 688 lake polygons in the GLOBathy dataset (Khazaei et al, 2022) using their mean and maximum depths ($V_d = 3\ d_{mean}/d_{max}$). Per grid cell, the lake volume is calculated as a 'reversed wedding cake' by multiplying the thickness of every discrete lake layer of the lake model with the average area at the layer depth A(z), calculated following Johansson et al., (2007) as

$$A(z) = A_{max}\left[\left(1 - d_{rel}\right)\left(1 + d_{rel}\sin\left(\sqrt{d_{rel}}\right)\right)\right]^{f(Vd)} \tag{2}$$

and

$$f(V_d) = 1.7 V_d^{-1} + 2.5 - 2.4\ V_d + 0.23\ V_d^3 \tag{3}$$





with $A_{max}$ (m²) the surface lake area calculated based on the gridded HydroLAKES distribution, $d_{rel}$ (m) the relative lake layer

depth ( $d_{rel} = z/z_{max}$), where $z_{max}$ (m) is given by the GLDB lake depth, and finally $V_d$ (-) the volume development parameter,

taken                                     constant                                     at                                     1.19.

Then, the lake heat content of every grid cell is calculated by combining the volume of every lake layer with the

layer temperature and integrating over the whole lake column. Next, the heat storage is computed from the globally

aggregated lake heat content values relative to the year 1960. To estimate heat uptake by reservoirs we account not only for

warming temperatures, but also include the increase in water volume through reservoir construction by using transient

reservoir area, in which reservoirs appear in their year of construction given by GRanD (Vanderkelen et al., 2021; 2022).

Finally, heat flux estimates are derived from the heat content time series by calculating the difference in heat content

between two consecutive years, divided by the total lake and reservoir area for the corresponding years.

**3 Results**

Estimates of ground heat flux by the new bootstrapping technique described above present slightly smaller values and a

narrower uncertainty range than the results from previous estimates using the Xibalbá dataset (von Schuckmann et al., 2020;

Cuesta-Valero et al., 2021a). Previous analyses present a global heat flux of 97±6 mW m$^{-2}$ for 1960-2018, in comparison

with 84.8±0.8 mW m$^{-2}$ for the period 1960-2020 in this study (Figure 1a). Both heat flux estimates consider the same

subsurface temperature profiles and the same singular value decomposition algorithm to produce inversions of individual

logs. Nevertheless, the new bootstrap method used to aggregate inversions from individual profiles is conceptually different

from the aggregation method used in von Schuckmann et al. (2020), which leads to slightly different values of global ground

heat flux and to a narrower 95 % confidence interval (Cuesta-Valero et al., 2022). The singular value decomposition method

in von Schuckmann et al. (2020) is based on deriving two extremal inversions from each individual profile, and then

considering the global uncertainty as the average of these extremal inversions. The bootstrap approach, nevertheless, derives

a set of 1000 different global averages from individual profiles considering a different quasi-equilibrium profile and a

different thermal diffusivity each time, retrieving the 2.5[th], 50[th], and 97.5[th] percentiles of these global averages. A more

detailed comparison between these two techniques can be consulted in Cuesta-Valero et al. (2022). The ground heat fluxes

from the bootstrap inversion technique are also higher than those from Beltrami et al. (2002), which presented 39.1 mW m$^{-2}$

for 1950-2000. These large differences between our results and those from Beltrami et al. (2002) arise from the use of

different inversion methods and from the higher number of more recent subsurface temperature profiles in the Xibalbá

dataset than in Beltrami et al. (2002), thus including the recent warming of the continental subsurface. Heat flux for inland

water bodies reaches 16±27 mW m$^{-2}$ for 1960-2020, displaying a large inter-annual variability and multimodel spread

(Figure 1a). This large inter-annual variability is explained by the smaller surface of global lakes and reservoirs in





comparison with the global land and permafrost areas. Permafrost heat flux estimates for the Arctic region yield 60±80 mW
       m$^{-2}$ for the period 1960-2020, thus lower than the ground and higher than inland water bodies during the same period of time.
       All three components present positive heat flux trends, with ground heat flux presenting a trend of 1.7 mW m$^{-2}$ y$^{-1}$, inland
       water bodies showing a trend of 1.3 mW m$^{-2}$ y$^{-1}$, and the trend for permafrost heat flux amounting to 0.9 mW m$^{-2}$ y$^{-1}$. Ground
       heat flux data after the year 2000 are an extrapolation of the 1970-2000 trend, which could imply an underestimation of the
trend for the whole period due to the fast change in global surface temperatures in recent times (Gulev et al., 2021).

       Estimates of heat storage per unit of area show large differences in the capacity to gain heat of each subsystem
       (Figure 1b), with the ground displaying a heat storage of 161.9±0.7 MJ m$^{-2}$ (1 MJ = 10$^6$ J), inland water bodies a heat gain of
       67±76 MJ m$^{-2}$, and permafrost soils a heat gain of 115±56 MJ m$^{-2}$ at the end of the period 1960-2020 (Figure 1b). There are
       also spatial differences in the retrieved heat storage per unit of area, with a general heat gain in inland water bodies and
subsurface temperature profiles around the globe, but with most permafrost heat gains arising from southern Arctic latitudes
       (Figure 2). Subsurface temperature profiles show a general increase of heat content in the ground, although with individual
       logs displaying heat losses at certain locations (Figure 2a). However, individual profiles are sensitive to microclimate
       conditions (e.g., Taylor et al., 2008), thus signals at individual locations may vary in comparison with the regional pattern.
       Regional differences appear in the heat storage per unit of area in inland water bodies (Figure 2b), showing a general heat
gain except in southeast Asia and around the Hudson Bay in Canada. Permafrost soils display small changes in heat content
       in northern Canada, northern Alaska and most of Siberia in contrast with a strong heat gain in the southern part of the Arctic
       region in North America and Asia (Figure 2c).

       The estimates of heat flux and heat storage per unit of area for inland water bodies are derived from the total heat
       storage for natural lakes and reservoirs, similar to Vanderkelen et al. (2020). These heat storage time series for natural lakes
represent the changing water temperatures, which show a positive trend from the 1990s onwards (Figure 3a). Our estimates,
       0.18 ± 0.19 ZJ for 2011 to 2020 relative to past times (1900-1929), are lower compared to previous estimates (0.29 ± 0.2 ZJ
       for the same period; Vanderkelen et al., 2020). This difference can be attributed to the additional simulations with the global
       lake model GOTM and the refined volume estimates. Contrary to the other simulations, the GOTM simulations forced by
       HadGEM2-ES and MIROC5 do not show an upward trend (Supplementary Figure S1). Using the $V_d$ parameter as a measure
for lake morphometry to calculate lake layer volumes, results in lower volumes compared to the cylindrical approach. A
       sensitivity analysis comparing heat storage for natural lakes with different global mean $V_d$ values and the cylindrical
       approach shows that heat storage increases with increasing $V_d$ values, while the cylindrical bathymetry results in distinct
       larger values (Supplementary Figure S2). This can be explained by the different lake volumes which are higher for concave
       shaped bathymetries ($V_d>1$) compared to more convex shaped bathymetries ($V_d<1$; Johansson et al., 2007). The cylindrical
approach results in the highest lake volumes and therefore the largest heat storage. Reservoir heat storage is an order of
       magnitude smaller compared to natural lakes with estimates of 0.21 ± 0.17 ZJ for 2011 to 2020, relative to past times (Figure
       3b). The steady increase originates from both reservoir construction, which accelerated in the years 1950-1970 and the
       increasing water temperatures. Finally, Vanderkelen et al. (2020) reports river heat storage estimates of -0.36 ± 1.2 ZJ for





2011 to 2020, relative to past times, based on water storage simulations by two global hydrological models within the
ISIMIP2b framework and river temperatures derived from surface temperatures of the GCMs. These estimates are
characterised by a large uncertainty, which is originating from a high variability in water storage, masking the positive
temperature trend (Vanderkelen et al., 2020).

The total continental heat storage since 1960 reaches 23.9±0.4 ZJ (1 ZJ = $10^{21}$ J), and is distributed over the
different components as follows: 21.6±0.2 ZJ is stored in the ground, 0.2±0.4 ZJ is stored in inland water bodies, and 2±2 ZJ
is used to thaw permafrost. This value of continental heat storage including the storage in the ground, water bodies and
permafrost thawing is, nevertheless, similar to the previous value of ~24 ZJ published in von Schuckmann et al. (2020) for
the period 1960-2018. This new estimate is within the 95 % confidence interval provided in von Schuckmann et al. (2020),
with the smaller values of ground heat flux in comparison with previous estimates explaining the lower values of continental
heat storage (see above). Ground heat storage accounts for the majority of continental heat, representing more than 90 % of
the continental heat storage for the period 1960-2020 (Figure 4). Inland water bodies store ~0.7 % of the total continental
heat, and permafrost thawing accounts for approximately 9 %. Nevertheless, our estimates of permafrost heat storage do not
include the thawing of ground ice in the Tibetan Plateau, thus the percentage corresponding to permafrost in Figure 4 is
probably larger than the value presented here.

## 4 Implications for Society and Ecosystems

Global climate models project a warming of the Earth system in the near future, even under low emission scenarios
(Tokarska et al., 2020; IPCC, 2021). These projections imply an amplification of the impacts associated with increases in
continental heat storage on society and ecosystems (Figure 5). Energy exchanges between the lower atmosphere and the
shallow subsurface determine the energy balance at the land surface, which connects the changes in net radiation, sensible
heat flux, latent heat flux, and ground heat flux (Bonan, 2002). As part of the land surface energy balance, and despite being
the smallest term in most situations (e.g., Bonan, 2002; Purdy et al., 2016), ground heat flux needs to be determined in order
to close the energy balance at the surface and minimise uncertainties in the rest of components. A complete knowledge of the
surface energy balance, together with soil conditions, is fundamental to understand the evolution of land-atmosphere
interactions affecting important climate and meteorological phenomena, such as surface temperature increase, surface
temperature variability, and extreme temperature events (Seneviratne et al., 2006; Fischer et al., 2007; Seneviratne et al.,
2013; Thiery et al., 2017; Vogel et al., 2017; Ma et al., 2018; Wang et al., 2022; Parmesan et al., 2022).

Increases in ground heat storage also produce a warmer subsurface, which threatens the stability of the soil carbon
pool by enhancing heterotrophic soil respiration and permafrost thawing, thus increasing emissions of greenhouse gases such
as carbon dioxide and methane, particularly from northern soils (Koven et al., 2011; MacDougall et al., 2012; Schädel et al.,
2014; Schuur et al., 2015; Hicks Pries et al., 2017; McGuire et al., 2018). Although permafrost heat storage is just 9 % of the
continental heat storage, the permafrost carbon feedback associated with permafrost degradation will add additional





greenhouse gases into the atmosphere at a pace of 18 PgC per Celsius degree of global warming by 2100 according to the IPCC 6th Assessment Report (Canadell et al., 2021), affecting the fulfilling of the temperature targets of the 2015 Paris Agreement (Natali et al., 2021). Furthermore, the risk of sudden thawing for carbon-rich zones in the Arctic subsurface, like the Yedoma region and peatland-rich regions, has increased in recent decades (Strauss et al., 2013; Nitzbon et al., 2020;

Fewster et al., 2022). The abrupt thaw of ground ice may constitute a tipping point for the climate system, mostly due to the release of carbon dioxide and methane and the long lifetime of carbon dioxide in the atmosphere (Lenton, 2012; Turetsky et al., 2019). Although the IPCC Special Report on the Ocean and Cryosphere in a Changing Climate indicates low to medium confidence in trespassing this tipping point in the 21st century (Collins et al., 2019), the consequences of crossing this dangerous threshold for the Earth System could be severe (Lenton et al., 2019). Health of northern communities can also be

affected by the degradation of this previously stable frozen layer, as contaminants such as radon can be released into the local freshwater systems (Furgal & Seguin, 2006; Cochand et al., 2019; Teufel & Sushama, 2019; Ji et al., 2021; Miner et al., 2021; Mohammed et al., 2021; Berry & Schnitter, 2022; Glover & Blouin, 2022). Permafrost thawing also alters the land, hampers travelling (Gädeke et al., 2021), and modifies traditional construction ways and sites at northern latitudes, damaging the mental health of northern communities (Lebel et al., 2022) and threatening industrial structures to retrieve

natural resources (Buslaev et al., 2021).

Heat uptake by inland water bodies and the associated increase in water temperatures are causing changes in lake ice cover duration and lake stratification, ultimately changing the thermal habitats of organisms (Woolway et al., 2020; Grant et al., 2021; Kraemer et al., 2021; Woolway et al.,2021b). These changes in the thermal state of inland freshwater systems are affecting ecosystem dynamics by degrading water quality, altering the carbon cycle, and producing algal blooms that

alter oxygen concentrations and primary productivity, which in turn threaten the food security of communities relaying in freshwater fisheries and other ecosystem services, like recreational activities (McIntyre et al., 2016; Woolway et al., 2020; Woolway et al., 2021a; Parmesan et al., 2022).

Therefore, it is clear that all three components of the continental heat storage are relevant for understanding the implications of climate change, independently of the different levels of heat storage estimated here. An analogous situation

arises from the analysis of the global Earth heat inventory (von Schuckmann et al., 2020), where the ocean is the leading reservoir of heat accounting for ~89 % of the total heat gain in the Earth System. However, changes in heat storage in the continental landmasses, the atmosphere, and the cryosphere are also important due to the associated repercussions for society and ecosystems. For instance, changes in cryosphere heat content account for just 4 % of the total heat gain in the system, but accurately quantifying future heat increases in this climate subsystem is critical to project sea level rise. In the same way,

permafrost heat storage may be just 9 % of the continental heat storage, but thawing of subsurface ice is a potentially large source of greenhouse gases due to its associated permafrost carbon feedback (Miner et al., 2022). Therefore, it is important to monitor all three components of the continental heat storage.



## 5 Conclusions

Continental heat storage has been estimated here considering inland water bodies and permafrost thawing in addition to the land subsurface for the first time. All three components present heat gains during the period 1960-2020, with total heat storage increasing more in the last decades of this period (Table 2). Determining the continental heat storage from all land components is important to accurately quantify the Earth heat inventory, the critical magnitude informing about future warming and climate change (von Schuckmann et al., 2020), as well as to understand the future consequences for society and ecosystems associated to continental heat gains (Figure 5). Monitoring the evolution of continental heat storage and its three subsystems is, therefore, important, and periodic updates of this analysis are planned with a frequency of 2-3 years in order to incorporate new data and techniques.

Certain aspects of the analysis presented here should be improved in next iterations. New measurements of subsurface temperature profiles are crucial to provide ground heat flux estimates for the last two decades of the period of interest. Values of ground heat flux from 2000 to 2020 in this analysis consist of an extrapolation due to the lack of sufficient profiles measured after the year 2000. Also, new measurements in the Southern Hemisphere and the Middle East are necessary to characterise areas without coverage in the global network of subsurface temperature profiles. Ideally, an international organisation should gather a copy of all available measured subsurface temperature profiles to ensure the maintenance and accessibility of these valuable records in future decades. Such a safe copy of all logs should lead to less fragmented datasets, harmonising the archiving practices and metadata requirements for all records, in contrast with current practices in which individual researchers are responsible for measuring, curating, storing, and distributing the data.

Several limitations are also present in our estimate of permafrost heat storage. The primary source of uncertainty in this analysis is the lack of accurate information about the amount and distribution of ground ice in permafrost regions. Additionally, the Tibetan Plateau, the Alpine regions and the southern hemisphere are not included in the analysis, thus the heat storage by melting of ground ice is probably slightly larger than the values presented here. Since it can be assumed that there is substantially more ground ice in the Arctic region than in the other regions (Zhang et al., 2008), only a small portion of the permafrost heat reservoir is likely to be missing. Among the limitations of the permafrost model are neglected modes of permafrost thaw such as thermokarst and thermo erosion. Furthermore, the model does not represent ground subsidence, a dynamic ground hydrology, and processes occurring on subgrid resolution. The absence of these processes affects the representation of the insulating capacity of the active layer thickness and likely leads to an underestimation of permafrost thaw (Lee et al., 2014; Rodenhizer et al., 2020, Smith et al., 2022a). Most of these limitations arise from the need to perform long-term simulations of permafrost evolution. Thereby the computational effort, the availability of input data and the process representation have to be balanced. Including the Tibetan Plateau and the permafrost zones of Antarctica will be possible in the near future, as those require a modest increase in computational resources and input data to derive soil stratigraphies.





Estimates of continental heat storage can potentially be used to constrain and evaluate transient climate simulations performed by global climate models. The Earth heat inventory has already being used to evaluate Historical simulations from the fifth phase of the Coupled Model Intercomparison Project (CMIP5), showing that these models present issues for representing a realistic distribution of stored heat among the different climate subsystems, as well as some energy conservation issues (Cuesta-Valero et al., 2021c). The same analysis indicates that the shallow continental subsurface

represented in the used Land Surface Model (LSM) components is one of the main reasons for their biassed representation of the heat inventory. Such result is in agreement with previous analyses comparing ground heat flux and ground heat storage from subsurface temperature profiles and climate simulations, which has lead to the development of deeper LSMs (MacDougall et al., 2008, 2010; Cuesta-Valero et al., 2016). Furthermore, this deeper subsurface in LSMs has also improved the representation of permafrost dynamics, showing how the ground heat storage retrieved from measurements of subsurface

temperature profiles have informed the development of climate models (Alexeev et al., 2007; Nicolsky et al., 2007; Hermoso de Mendoza et al., 2020; González-Rouco et al., 2021; Steinert et al., 2021). Another approach may be to use the retrieved estimates of continental heat storage as a reference to constraint projections of climate change (Tokarska et al., 2020; Ribes et al., 2021). That is, climate models could be classified depending on how well the models reproduce the change in heat storage in the different components, as it is done with surface temperature increases and other variables (Schmidt et al.,

2014; Harrison et al., 2015; Eyring et al., 2019). However, the potential of continental heat storage as a reference may be hampered by the use of models to determine the evolution of heat storage in permafrost soils and inland water bodies, as observations are preferred for evaluating climate simulations.

There are several steps that can be implemented for improving future estimates of ground heat storage. Expanding the number of estimates of ground heat flux has a high priority, as the ground heat storage is the largest term of the

continental heat storage. New measurements of subsurface temperature profiles in areas not well represented in the current global database, regions such as northern and central Africa, South America, the Middle East, and southeastern Asia, are important to improve the spatial coverage of the current subsurface temperature dataset. Furthermore, strengthening the global network of subsurface profiles by repeating measurements at previously measured sites, would reduce uncertainties for the warming of the continental subsurface in the last decades. Flux estimates from other datasets can be also considered,

such as from FluxNet towers and from satellite remote sensing data. Indeed, there is an increasing population of satellites providing information about land surface conditions and changes in land cover, and several methods are also being developed to obtain accurate estimates of climate variables from satellite remote sensing observations in combination with land observational networks and numerical models (Balsamo et al., 2018).

Expanding the permafrost areas considered here would be a priority in the next iterations of this analysis,

particularly the inclusion of the Tibetan Plateau. Further sources of information to retrieve estimates of permafrost heat storage should also be considered in order to increase the confidence of the results obtained here. Ideally, monitoring of liquid water content in permafrost soils along with borehole temperature measurements would form a complementary dataset besides modelling for estimating permafrost heat storage. However, the current observational networks in the Arctic and on





the Tibetan Plateau are not equipped to perform such measurements, and their spatial coverage should be expanded to
include currently unmeasured zones in the Canadian Arctic and Eurasia (Biskaborn et al., 2015). Multimodel simulations
using land surface models that represent permafrost such as those from the sixth phase of the Coupled Model
Intercomparison Project (CMIP6) may be considered for including the uncertainty due to model structure in the analysis.
However, direct use of the simulated ice content and soil temperatures from the CMIP simulations is not currently possible
because the representation of the soil in the models is too shallow to assess the evolution of the thermal state of the ground
beyond the near-surface permafrost (Koven et al., 2013; Slater & Lawrence, 2013; Burke et al., 2020; Hermoso de Mendoza
et al., 2020; Steinert et al., 2021). Furthermore, these land surface model components do not represent excess ice in the
ground or ground subsidence, which bias the represented permafrost thawing (e.g., Lee et al., 2014; Rodenhizer et al., 2020).
Replacing ERA-Interim forcing by ERA5 data and forcing for the last decades of the 20[th] century and the first decades of the
21[st] century should also be considered.

395       The inland water heat storage estimates could be refined using spatially varying morphometry characteristics to
determine lake volumes per grid cell. The availability of new datasets like GLOBathy (Khazaei et al., 2022), as well as new
insights from the upcoming Surface Water and Ocean Topography (SWOT) mission would allow such an approach. Using
this lake morphology together with an updated lake mask in the global lake model simulations would further improve the
lake temperature trends. Such simulations will become available in the upcoming ISIMIP3 round (Golub et al., 2022). In
addition, emerging remote sensing products of lake surface temperatures can be used to better calibrate and validate the
global lake models, which will further improve the simulated temperature profiles (Golub et al., 2022). Finally, to improve
the estimates of heat stored in rivers, better estimates of the water volumes in rivers are required, in addition to explicitly
modelled river temperatures (Wanders et al., 2019). These will be included in ISIMIP3, as process-based global hydrological
models now also simulate river temperatures.


**Data and code availability**

The subsurface temperature profiles from the Xibalbá dataset were used to derive ground heat fluxes and are available in
Cuesta-Valero et al. (2021b). All ISIMIP2b global lake simulations used are publicly available through the ISIMIP
repository (https://data.isimip.org/). The HydroLAKES dataset is available at https://www.hydrosheds.org/page/hydrolakes,
GRanD at http://globaldamwatch.org/ and GLDB at http://www.lakemodel.net. The scripts used for the inland water heat
storage calculations are available at: https://github.com/Ivanderkelen/inlandwater_heatuptake.



**Author contributions**

Overall coordination of this initiative has been driven by F.J. Cuesta-Valero and K. von Schuckmann. Estimates of ground heat flux and ground heat storage were provided by F. J. Cuesta-Valero. Estimates of permafrost heat storage were provided by J. Nitzbon and M. Langer. Estimates of heat flux and heat storage from inland water bodies were provided by I. Vanderkelen and W. Thiery. All authors contributed to the analysis of results and the list of implications. F.J. Cuesta-Valero wrote the manuscript with continuous input from all authors.

**Funding information**

- Francisco José Cuesta-Valero is an Alexander von Humboldt Research Fellow at the Centre for Environmental Research (UFZ).
- Hugo Beltrami was supported by grants from the National Sciences and Engineering Research Council of Canada Discovery Grant (NSERC DG 140576948), the Canada Research Chairs Program (CRC 230687), the Canadian Foundation for Innovation and the Digital Research Alliance of Canada (Compute Canada, AceNet). Hugo Beltrami holds a Canada Research Chair in Climate Dynamics.
- Moritz Langer and Jan Nitzbon were supported by a grant of the German Federal Ministry of Education and Research (BMBF, project PermaRisk, grant no. 01LN1709A).
- Inne Vanderkelen is a research fellow at the Research Foundation Flanders (grant no. FWOTM920). The resources and services used in this work were provided by the VSC (Flemish Supercomputer Center), funded by the Research Foundation - Flanders (FWO) and the Flemish Government.





**Tables**

**Table 1:** Overview of global lake models used. Detailed descriptions of the models and modelling setup can be found in Golub et al. (2022).

| Lake model | Number of layers | Lake depth | Reference |
|---|---|---|---|
| CLM4.5 | 10 | Constant at 50 m | Subin et al. (2012) |
| SIMSTRAT-UoG | 1 - 13* | GLDB v1 | Goutdsmit et al. (2002) |
| ALBM | 51 | GLDB v1 | Tan et al. (2015) |
| GOTM | 10 | GLDB v1 | Sachse et al. (2014) |

*\* The number of lake layers used in SIMSTRAT-UoG varies spatially and depends on the mean lake depth of the grid cell.*

**Table 2:** Continental heat storage (CHS) from this analysis and from von Schuckmann et al. (2020) (vS20) in ZJ (1 ZJ = $10^{21}$ J). Results for ground heat storage (GHS), permafrost heat storage (PHS), and inland water heat storage (IWHS) are also displayed.

| | vS20 | CHS | GHC | PHS | IWHS |
|---|---|---|---|---|---|
| 2010-2020 | 21.5±1.7 | 20.97±0.14 | 18.83±0.05 | 2.0±0.6 | 0.17±0.13 |
| 2000-2010 | 16.3±1.4 | 15.32±0.11 | 13.74±0.04 | 1.5±0.4 | 0.11±0.10 |
| 1990-2000 | 11.2±1.2 | 10.43±0.07 | 9.398±0.029 | 0.99±0.29 | 0.04±0.06 |
| 1980-1990 | 6.8±1.0 | 6.54±0.06 | 5.926±0.021 | 0.61±0.18 | 0.00±0.06 |
| 1970-1980 | 3.3±0.6 | 3.52±0.06 | 3.228±0.015 | 0.30±0.09 | -0.01±0.06 |
| 1960-1970 | 0.87±0.27 | 1.05±0.05 | 1.007±0.007 | 0.058±0.029 | -0.02±0.05 |





**Figures**

**Figure 1:** Global heat flux (top panel) and global heat storage per unit of area (bottom panel) from the ground (red lines), inland water bodies (blue line) and permafrost thawing (green line) for the period 1960-2020. Black lines indicate ground heat fluxes and ground heat storage from von Schuckmann et al. (2020). Please, note that ground estimates consist in long-term changes of heat flux and heat storage, and do not include inter-annual variability.






**Figure 2:** Spatial distribution of heat storage per unit of area since 1960 for (a) ground heat storage from subsurface temperature profiles measured after 1990, (b) heat storage from inland water bodies, and (c) heat storage from permafrost thawing. Please, note the different scale for permafrost heat storage.

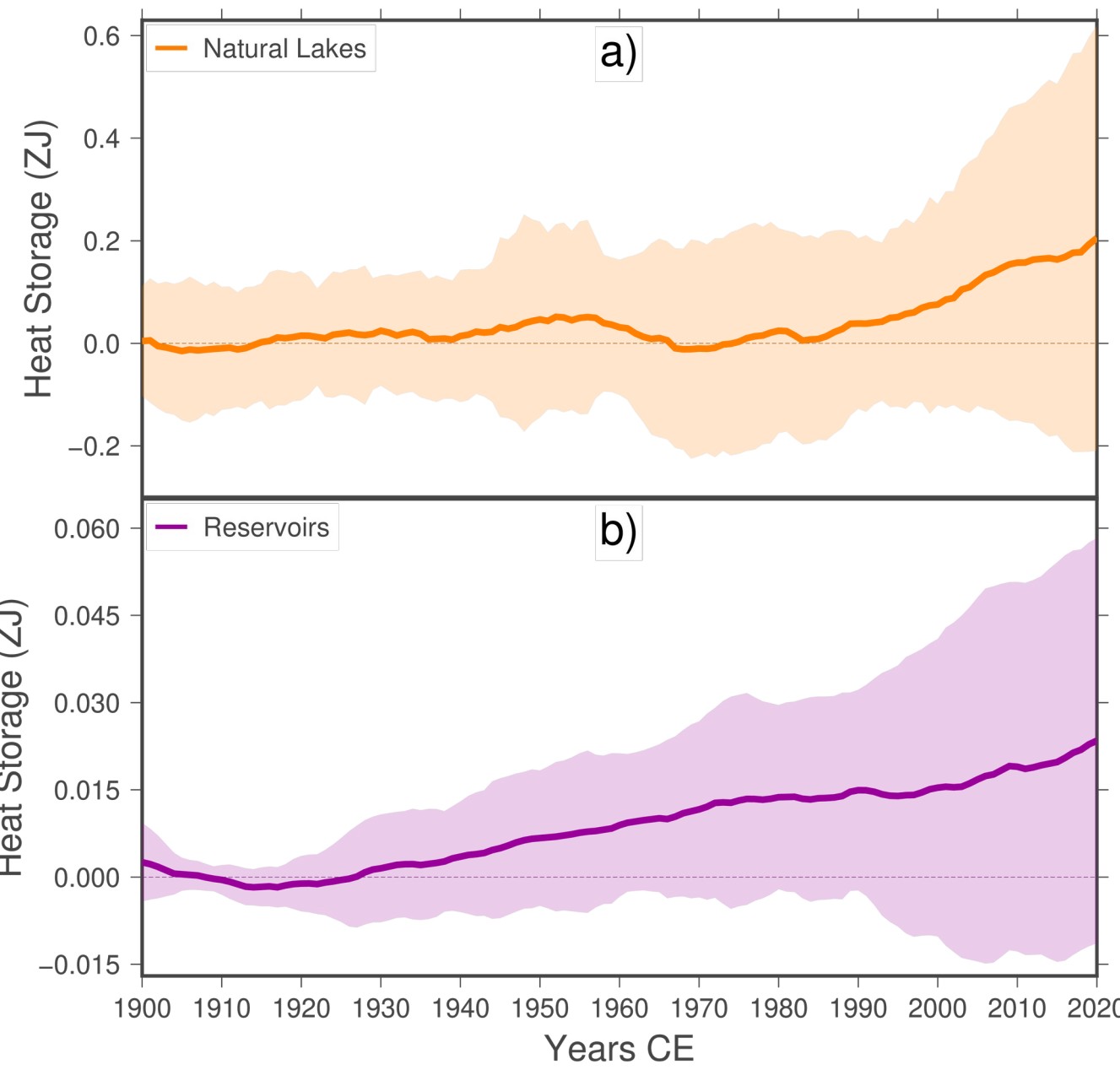

**Figure 3:** Heat storage by natural lakes (a), reservoirs (b). Shown are 6-year moving averages relative to the 1900–1929 reference period. Note the different y-axis scales. Colour shades represent uncertainty range shown as the standard deviation of the used simulations (16 for lake and reservoir heat storage).





**Figure 4:** Percentage (%) of the continental heat storage within each analysed land component for the period 1960-2020: ground (red),
inland water bodies (blue), and permafrost degradation (green). Left axis corresponds to ground results, the right axis corresponds to
results for permafrost soils and inland water bodies.





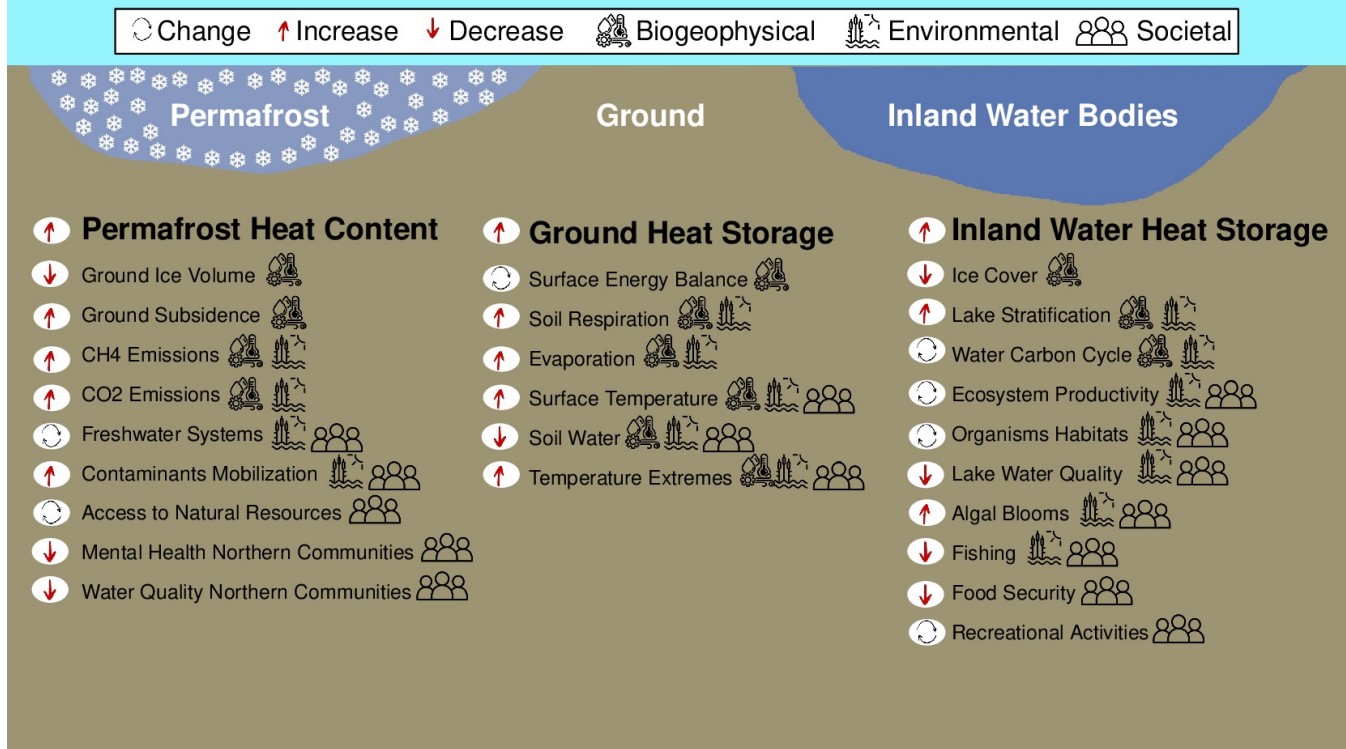

**Figure 5**: Environmental processes and societal implications affected by changes in heat storage within the continental subsurface (ground), inland water bodies, and permafrost soils. Arrows indicate the direction in change for each process according to the increases in heat storage in the corresponding component of the continental heat storage. See Section 4 for more details.





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
