# Peer review of "Continental Heat Storage: Contributions from the Ground, Inland Waters, and Permafrost Thawing"

_Earth System Dynamics, 2022_

## Author Comment (AC1)

*Dear Reviewer,*

*We thank you for your thorough and constructive feedback. This file provides a complete documentation of the changes made in response to each of your comments. Reviewer's comments are shown in normal text, author responses are shown in bold, italic, blue text.*

**Reviewer 1**

General Comments

The manuscript submitted by Cuesta-Valero et al. considers continental heat storage and determines the contribution from three components. The analysis is important as it contributes to better understanding of the overall global heat balance by ensuring that all components are accounted for in the calculation of continental heat storage. The subject area is therefore appropriate for publication in ESD and would be of interest to its readers. The MS is also relevant to better estimates of the impact of climate change on the landmass. The MS has clear objectives and is generally well written with results and interpretations presented clearly. I don't have any major concerns with the MS but I do have a number of comments that should be considered prior to acceptance for publication.

One of the key things that is done in the paper is the calculation of the heat in the ground that is utilized for phase change (latent heat) as ice in permafrost melts. However, the way the paper is written the authors seem to consider this separate from the subsurface (or ground) heat storage, which I found odd. Permafrost is a component of the ground (essentially a thermal condition of the ground) in cold environments so both the heat used to raise its temperature or for phase change when it thaws are components of the heat that is stored in the ground. It would seem that this is more an issue of the method that has been traditionally utilized to determine ground heat storage. Analysis utilizing subsurface temperature profiles only considers conduction in the estimate of ground heat fluxes. As ground temperatures approach 0 °C in permafrost, heat is utilized for phase change of any ice in the ground rather than raising the temperature and little change in temperature over time is observed in ground temperature profiles (as discussed in Romanovksy et al. 2010; Smith et al. 2010). Lack of consideration of the latent heat effects therefore means that ground heat storage determined considering only conduction would be underestimated. It would make more sense for the authors to say that they are refining the estimates of ground heat storage by addressing a limitation of the method traditionally used by considering the latent heat utilized for phase change in the estimates.

*The reviewer is right that permafrost is just perennially frozen ground and that our permafrost heat storage estimate is essentially the change in latent heat storage. Furthermore, available methods to estimate ground heat storage from subsurface temperature profiles cannot include latent heat flux used to thaw permafrost, as indicated by the reviewer.*

*In our analysis, we use a model and a series of assumptions about the stratigraphies of the Arctic subsurface in order to estimate the latent heat used in permafrost thawing, in order to complement the observation-based method used to derive sensible ground heat storage. That is, we separate the sensible and latent heat fluxes, mainly due to methodological limitations. Therefore, we believe that it is better if we maintain both estimates of heat storage as separate entities in order to improve the clarity of the manuscript.*

*We have added a couple of lines in the new version of the manuscript to make clear the division into sensible and latent heat fluxes.*

The authors do not mention the role of other modes of heat flux in the ground such as convection. Heat transfer associated with water movement (advection) such as infiltration of precipitation and snow melt or subsurface water flow may also influence the ground thermal regime (see for eg. Douglas et al. 2020; Neumann et al. 2019; Phillips et al. 2016; review of Smith et al. 2022b also discusses this). As permafrost thawing occurs, subsurface water flow becomes more important. Is lack of consideration of this mechanism of heat flow also a limitation of the method used to determine ground heat storage?

*Indeed, our approach to derive estimates of permafrost heat storage is not able to include an active hydrology in the subsurface. We have noted this fact as a limitation in the manuscript.*

*Regarding advection in subsurface temperature profiles, the diameter of the drilling holes is usually small enough to prevent air advection. Water advection is still possible, which may introduce a non-climatic signal in the profiles. Nevertheless, all logs were screened by eye, and those including signals that cannot be explained by climate alone were removed (CuestaValero2021ghcbore).*

*We have added a couple of lines in the new version of the manuscript clarifying this point.*

I have a number of additional comments (see below) for the authors' consideration in preparing the revised manuscript. These comments identify where further clarification or information may be required. Suggestions for editorial revisions have also been provided.

Specific comments (keyed to line number)

L31 – See comment above – permafrost is the ground (earth material) so its thaw is a component of subsurface heat storage.

*We have already addressed this comment above.*

L32 – Suggested revision: " The ground accounts for 90

*Done.*

L41 – What is included in "cryosphere"? Permafrost is a component of the cryosphere but it is treated separately in this paper.

*In this context, the term cryosphere refers to glaciers and ice caps. We have indicated this in the new version of the manuscript.*

L53 – Permafrost includes soil and rock. Since there can be water within rock, phase change can also occur in frozen rock (even if the amount is small compared to soils).

*We have changed the text to reflect this point.*

L55 – replace "underline" with "underlie"

*Done.*

L55 – Note Obu (2021) determines the equilibrium permafrost distribution so it does not consider permafrost that formed under a colder climate and still persists today. For example, permafrost in peatlands in the southern portion of the permafrost regions formed under colder conditions and is preserved due to the insulating properties of peat. Also, permafrost can be quite thick in the Arctic and it can take a century or more to completely thaw so that relict permafrost continues to exist as climate warms.

*This is correct. We wanted to give an idea of the total area underlain by permafrost. Please note that the reported warming for permafrost after the* Obu2021globalpropperma *reference corresponds to recent times.*

L56 – It is important to note that these are average values of warming based on several sites (I believe Biskaborn 2019 gives a range).

*We have added the uncertainty ranges to the new version of the manuscript.*

L59 – Misleading/incorrect statement. These simulations only consider the upper 2-3m of permafrost

rather than its total vertical extent, which may be 10s to 100s m. These values therefore do not refer to complete loss of permafrost from this area (i.e. refer to thaw being more than 2-3m over this area).

*CMIP simulations from* koven2013analysis, slater2013diagnosing *and* burke2020permacmip6 *consider only shallow permafrost. Nevertheless, the LSMs considered in* hermoso2020bbclm4 *and in* Steinert2021lsmdepthhydro *consider soil columns with hundreds of meters of depth. However, we agree with the reviewer that the range of change in global permafrost extension refers to shallow permafrost, thus we have changed this in the new version of the paper.*

L61 – Permafrost is frozen ground so permafrost heat uptake is ground heat uptake. Until it thaws, the heat storage would be accounted for by the methods (inversion of temperature profiles) utilized to determine ground heat storage.

*We completely agree with the reviewer. Because of this, we refer only to the change in the area of permafrost in the previous line, and not to the change in permafrost temperature.*

L66 – What is meant by recent times? It would be clearer to give the time period over which this reduction occurred.

*We meant the last three decades. We have included this period on the text.*

L67 – suggested revision: ' ….going to continue throughout the 21st century…:

*Done.*

L79 – should this be "deep subsurface temperature profiles"

*Done.*

L87 – replace "in" with "of"

*Done.*

L89 – revise to "slope of this regression line" (or best-fit line)

*Done.*

L99-100 – If the time for temperature changes at the surface to reach a given depth depends on the thermal properties, how does truncating to the same depth yield the same temporal reference if thermal properties are variable?

*The reviewer is right, the time required for a surface perturbation to reach a certain depth depends on thermal properties. What we are assuming to provide the temporal reference indicated in line 101 is an homogeneous subsurface with a thermal diffusivity of $1.0 \times 10^{-6}$ m$^2$ s$^{-1}$, which is a typical value for bedrock. We have modified the text to clarify this point.*

L131-134 – I may have missed something here - how are the results from point-based measurements applied to the entire area considered. In figure 2a, heat storage is shown for points that are not uniformly distributed with very large areas not represented. It isn't clear how the point-based data are extrapolated to the larger area or what other information may be utilized especially give the large areas with no data.

*We followed the methodology in CuestaValero2021ghcbore, consisting in obtaining the averaged heat flux from all 1079 subsurface temperature profiles, and then estimating the accumulated heat considering a global land surface of $1.34 \times 10^{14}$ m$^2$. That is, we consider the area of all continents excluding Antarctica and Greenland, since we have no measurements there. This is possible because previous works have shown that the current distribution of boreholes is enough to capture global changes in surface conditions (pollack2004borehole; GarciaGarcia2016cmip5boreholes). Furthermore, CuestaValero2021ghcbore showed that changing the area considered does not affect the global estimates.*

*We have changed this paragraph in the new version of the manuscript to enhance the clarity of the text.*

L136 – Isn't it more correct to say that the heat input to the subsurface is utilized to melt ground ice as permafrost temperatures approaches 0 °C?

*Indeed, that is the complete physical process: permafrost thaws once the ground temperature is near zero Celsius degrees and the heat keeps getting into the ground. We have added a couple of lines in the text to explain the entire process.*

L140 – Do you mean the surface offset which is the difference between mean annual air and ground surface temperatures and is influenced by snow cover. The thermal offset refers to the difference in temperature between the ground surface and the top of permafrost, which (if equilibrium conditions exist) depends on difference between frozen and unfrozen thermal conductivity (See for e.g. Riseborough et al. 2008).

*We fully agree with the reviewer and changed the formulation accordingly.*

L143 – What about rock – permafrost includes rock which can contain ice.

*We used the dataset by Pelletier2016regolith to set the soil thickness and assumed bedrock underneath. The water/ice content in the bedrock was reduced compared to the overlying soil. Both parameters (soil thickness and bedrock ice content) were varied during the ensemble simulations to address the related uncertainties. Please see Langer2022cryogridlittle for details.*

L179 – How is depth determined?

*The lake depth is given by the Global Lake Database v.3(Choulga2019surfacewaterdepth), as indicated in line 176 of the original manuscript.*

L165-199 – Lakes can form or drain in the Arctic due to permafrost thaw. Is the change in surface water distribution due to thermokarst processes considered or is this a limitation to heat storage estimates?

*Unfortunately, the permafrost model used here cannot represent thermokarst processes nor water redistribution. We detailed those limitations in line 341 of the original manuscript. Furthermore, we did not consider dynamic (thermokarst) lake changes in the inland water heat storage which relied on a static lake distribution. We would like to note that the overall trend of thermokarst lake dynamics is very uncertain since both lake expansion and drainage happen concurrently. For the study period of the past few decades, the net lake area change is likely negligible compared to the total lake area.*

L220 (also elsewhere in paper including L223) – See earlier comments. Permafrost heat flux, if thaw is not is not occurring (this will be the case where temperature below melting point of ice in the ground) will be considered in the estimates of subsurface storage determined utilizing subsurface temperature records. It is only when thaw occurs in warmer permafrost at temperatures near 0 °C that latent heat needs to be considered in addition to conduction.

*Please, see comment about L136 above. We only consider permafrost heat flux as latent heat flux. Permafrost warming is only considered from subsurface temperature profiles. We have added a clarification in Section 2.2.*

L235 – Where around Hudson Bay? There was cooling in the eastern Arctic including northern Quebec into the 1990s – is this the reason for the lack of heat gain in this area?

*Indeed, there is a decrease in inland waters heat storage in the southwestern shore of the Hudson Bay. (Figure 2 of the original manuscript). Unfortunately, we are unable to explain this result, and we found no explanation in the literature either. We have reported this issue in the new version of the manuscript.*

L267 – Why isn't the Tibetan Plateau included given it is a fairly significant area. Permafrost in this region is generally warm so latent heat effects are important.

*Indeed, the Tibetan Plateau is an important region that should be included in the analysis. However, the simulated permafrost relied on an input dataset of soil organic carbon (hugelius2014permacarbon) which is only available for the northern permafrost region excluding the Tibetan plateau. It is planned to include the Tibetan Plateau in the next iteration of this analysis, as indicated in the manuscript.*

L275-276 – It is important to indicate here that the estimate of ground heat flux needs to consider non conductive heat flow (i.e. address limitations) to improve estimates. The MS makes progress in addressing this limitation by considering the latent heat associated with phase change as permafrost thaws.

*We think that advection is not significant for ground heat flux at the global scale. For example, huang20061851 uses meteorological observations of surface air temperature (SAT) to derive the evolution of global ground heat flux, reaching similar results to those in (beltrami2002earthmemory) from subsurface temperature profiles (GST). If nonconductive processes were relevant at the global scale, these two estimates should be different, as SAT observations would not account for these additional processes. We find that this result indicates that heat transport by conduction is the leading mode of heat diffusion trough the subsurface, with the exception of permafrost soils where thawing/freezing is occurring. Furthermore, Xibalbá profiles were screened to remove logs including advection (CuestaValero2021ghcbore), as indicated in the manuscript.*

L280-300 – This section is OK but most of this has been well covered in other publications so nothing really new here.

*Indeed, this part of the text is based on previous publications. Our aim was to reflect the most important results related to permafrost heat storage in order to provide a picture of the impacts that permafrost thawing posses for society and ecosystems. We have added also a small comparison with other components of the cryosphere in the new version of the manuscript in order to place our estimate in the context of the global ice budget.*

L280-285 –Other implications of ground warming and permafrost thaw are impacts on landscape processes and stability, changes to surface water distribution and increase in subsurface water flow. These impacts can also have feedbacks to the ground thermal regime with further impacts on carbon feedback.

*We have added these points in Section 4 of the new version of the manuscript.*

L288-290 – This is really an issue of landscape change associated with thawing of ice-rich permafrost (such as subsidence, thaw slumps), which is abrupt or sudden, exacerbating permafrost thaw – with geomorphic change such as slumps and other slope failures the upper boundary changes as material is removed (also lateral heat flow).

*Please, see the added text to answer the previous comment.*

L293 – Do you mean "surpassing" rather than "trespassing"

*Yes, we meant "surpassing". This is now fixed on the text.*

L295-300 – Other impacts related to permafrost thaw (especially if ice-rich) include loss of bearing strength and ground settlement/subsidence with impacts on infrastructure; landscape instability including slope failures which can release sediment into water bodies with implications for water quality; impacts on integrity of contaminant containment facilities.

*Please check Section 4 in the new version of the manuscript, we have noted those points in there.*

L301-303 – more evaporation?

*Indeed, global lakes have experienced larger evaporation rates in recent decades. Nevertheless, the leading factors causing this evaporation increase seem to be ice cover reductions (Wang2018evaporationlakes, Zhao2022evaporationlakes). However, for low latitude lakes, evaporation could increase by the process that lake surface temperatures warm at a slower rate than the overlying air, which leaves more energy from long-wave radiation available for lake evaporation (Wang2018evaporationlakes). We have indicated this in the new version of the text.*

L325-335 – There are several recent ground temperature records in the permafrost regions (some results included in Smith et al. 2022b, Noetzli et al. 2022, Biskaborn et al. 2019 and other papers). These are generally at shallower depths (usually upper 20 m) than would be utilized for the inversion of ground temperature profiles that is utilized in the MS. However, these provide information at depths where latent heat effects are important.

*We are aware of those ground temperature measurements, and we are planing to include them in a future iteration of this analysis. We have added some lines in the new version of the manuscript to clarify this point.*

L337 – This is not a new observation and the lack of ground ice information has been identified as a limitation in permafrost modelling in other papers (e.g. Smith et al. 2022b; O'Neill et al. 2020).

*Yes, this is not a new result. However, this is an important limitation affecting our results, thus we think that an explanation must be included in the text for completeness and transparency.*

L347 – With respect to latent heat effects related to permafrost thaw, including the Tibetan Plateau is probably more important than permafrost zones of Antarctica given the rather dry conditions and the geology.

*Correct, and because of that we plan to include the Tibetan Plateau as soon as computational and financial resources are available, moving towards achieving global coverage in a later iteration.*

L358-359 – While the deeper subsurface is an improvement, the LSMs still have limitations with respect to representation of subsurface conditions including ground ice distribution.

*Indeed, the lack of accurate data about the distribution of ground ice affects model development, as well as other research fields. But beyond ice representation, the depth of the LSM also affects subsurface thermodynamics, and in this regard the expansion of the LSMs' depth has improved the simulated permafrost in global climate models (nicolsky2007bbcperma).*

L382 – As mentioned in previous comment there are borehole temperature measurements in permafrost and at some sites, there are moisture content measurements. There are also often observations of excess ice content when boreholes are drilled.

*Indeed, sometimes you can have some borehole sites with more complete measurements. But the problem is that those extended measurements are seldom available, and that their number is very reduced. Therefore, those sites are very probably not representing global conditions, nor have them a sufficient temporal coverage to include decadal changes in temperature. Such limitations make them, therefore, unsuitable for the scope of our analysis.*

L385 – One of the issues in areas such as the Canadian Arctic is the remoteness and significant cost of drilling boreholes, especially deeper ones where specialized equipment needs to be transported to the site (see for e.g. Smith et al. 2022b). Most permafrost monitoring sites therefore are often located near communities, existing infrastructure, associated with resource development (hydrocarbon, mining) etc.

*Exactly, permafrost monitoring is a complex task because of the difficulty for maintaining the sites and covering such a vast extension of land. We have included this point in the new version of the manuscript.*

L392 – This is also discussed in Smith et al. (2022b) and O'Neill et al. (2020). There are also efforts to

improve ground ice potential modelling and mapping – see for e.g. O'Neill et al. (2019)

*We have included this point in the new version of the manuscript.*

Figure 5 – See previous comments regarding other implications of permafrost thaw such as impacts on infrastructure integrity. Landscape instability is a more inclusive term than ground subsidence.

*We have replaced ground subsidence for landscape instability in Figure 5.*

References cited in comments

Douglas, T. A., Turetsky, M. R. & Koven, C. D. 2020. Increased rainfall stimulates permafrost thaw across a variety of Interior Alaskan boreal ecosystems. npj Clim. Atmos. Sci. 3, 28.

Neumann, R. B. et al. 2019. Warming effects of spring rainfall increase methane emissions from thawing permafrost. Geophys. Res. Lett. 46, 1393–1401.

Noetzli, J. et al. 2022. [Global Climate] Permafrost Thermal State [in "State of the Climate in 2022]; Bull. Am. Met. Soc. Supplement, 103 (8)

O'Neill HB, et al. (2020) Abrupt permafrost thaw and northern development: Comment on "Abrupt changes across the Arctic permafrost region endanger northern development" by B. Teufel and L. Sushama. Nature Climate Change 10:722-723

O'Neill, H. B., Wolfe, S. A. & Duchesne, C. 2019. New ground ice maps for Canada using a paleogeographic modelling approach. Cryosphere 13, 753–773. – See also O'Neill et al.

Phillips, M., et al. (2016). Seasonally intermittent water flow through deep fractures in an Alpine Rock Ridge: Gemsstock, Central Swiss Alps. Cold Regions Science and Technology, 125, 117–127. https://doi.org/10.101

Riseborough D, et al. (2008) Recent advances in permafrost modelling. Permafrost and Periglacial Processes 19 (2):137-156. doi:10.1002/ppp.615

Romanovsky VE, Smith SL, Christiansen HH (2010) Permafrost thermal state in the polar Northern Hemisphere during the International Polar Year 2007-2009: a synthesis. Permafrost and Periglacial Processes 21:106-116

Smith SL, Romanovsky VE, Lewkowicz AG, Burn CR, Allard M, Clow GD, Yoshikawa K, Throop J (2010) Thermal state of permafrost in North America - A contribution to the International Polar Year. Permafrost and Periglacial Processes 21:117-135. doi:10.1002/ppp.690

---

## Author Comment (AC2)

*Dear Reviewer,*

*We thank you for your thorough and constructive feedback. This file provides a complete documentation of the changes made in response to each of your comments. Reviewer's comments are shown in normal text, author responses are shown in bold, italic, blue text.*

**Reviewer 2**

Cuesta-Valero et al. provide a new estimate of continental heat storage including ground, inland waters and permafrost thawing. For continental heat storage, an update to the previous estimate (Cuesta-Valero et al. 2021) is provided. For inland waters and permafrost thawing, models are used to derive the estimates. I have some major reservations about their methodologies, listed below.

(1). The observation-based estimate for ground heat storage and model-based estimates for inland waters and permafrost thawing are merged together to provide the continental heat storage. I doubt if they can be put together, and then eventually be used in von Schuckmann et al. GCOS assessment (the other components are all observation-based).

*We respectfully disagree with the reviewer on this point. Many relevant studies have combined raw observations, data assimilation (i.e., reanalysis and satellite products), as well as numerical simulations (i.e, global and regional climate simulations) in order to assess the state and evolution of a certain variable of interest. The different Assessment Reports (ARs) of the Intergovernmental Panel on Climate Change (IPCC) are the most important examples of this practice in the climate community. A particular example could be the combination of paleoreconstructions and paleosimulations to obtain a better picture of the last millennium in both IPCC-AR5 (ipcc5chap5) and IPCC-AR6 (ipcc6ts).*

*Furthermore, the rest of components of the GCOS assessment of Earth heat inventory do not include only observational data. For example, the atmosphere heat storage is estimated from reanalysis data, while the heat uptake by glacier melting is estimated from indirect gravimetric observations retrieved from satellites. These two estimates are produced by using numerical techniques and models to assimilate and interpret raw observations. Similarly, we use an observational-based driver to force a numerical model to estimate permafrost heat storage. Concretely, we use ERA-Interim data as upper boundary condition for our permafrost model, which allows us to estimate the ground ice melting that is coherent with the evolution of surface conditions in the last decades. Please note*

*that this is not very different to the use of different reanalyses to estimate the heat storage by the atmosphere in the GCOS paper mentioned by the reviewer.*

*Finally, we must also note that the use of models in our estimates is mostly the result of a lack of adequate data to characterize heat storage in permafrost and inland waters systems. For lakes, for example, existing in-situ data sets with long-term temperature profiles contain only very few lakes relative to the total number of lakes worldwide, and existing repositories are spatially biased towards Europe and North America. For rivers, the data availability is even worse. As we make clear in the manuscript, we incorporate model results because, unfortunately, there are no adequate measurements to derive global estimates of permafrost heat storage and inland waters heat storage.*

*We have also added a new paragraph in the Conclusions stating that although this is not an ideal estimate of continental heat storage, we have identified a clear path toward complementing the use of models with more observational-based estimates.*

(2). Uncertainty estimates for ground heat storage. In this study the uncertainty of the ground heat storage has been reduced by an order compared to their earlier estimate (for example line 200-205). The new estimate suggests a global land heat storage of 84.8 +/- 0.8 mWm-2 (previous estimate is 97+/-6). I found it hard to believe such a small error range, it is simply not possible. Remember you are using only 1000 station data to represent the entire land, even previous error range of 6 is a likely underestimation. I can't understand this small number and I don't understand how this small number is derived given the dataset is basically the same with the previous version.

*Please, note that ground heat storage is estimated from subsurface (borehole) temperature profiles, not from meteorological stations. These temperature-depth profiles record the propagation of alterations in the surface energy balance though the ground, but due to the nature of heat diffusion, borehole profiles are able to retrieve only long-term past changes in surface conditions. That is, decadal to centennial changes in ground heat flux. This reduces greatly the variability in the global average of ground heat storage in our manuscript, even considering the variability at 1079 different locations, thus uncertainty ranges are always going to be narrower than those of estimates based on meteorological stations.*

*Furthermore, we have used a new bootstrap technique to estimate uncertainties from these geothermal data. Previous estimates of global ground heat flux from subsurface temperature profiles provided uncertainty estimates that were biased from a statistical point of view, as they were markedly conservative and included a much wider range than the 95 % confidence interval that is typically*

*provided with the global average. Please, check* cuesta-valero2022bti *for a detailed comparison of previous uncertainty estimates in comparison with the new bootstrap method, as well as a prove that the uncertainty in previous inversions converge to the one reported here when appropriate error propagation methods are considered.*

(3). Uncertainty estimates for permafrost thawing. Only the uncertainty related to the soil thickness and ice saturation are taken into account. However, I think another major error come from the model and climate forcing. For example, the use of Mk3L and ERA-Interim, the errors/biases will definitely propagate into the estimate of this study. I have no idea how to resolve this, as it is related to the fundamental choices of this study: using models and reanalysis to drive the their estimates.

*Indeed, the use of numerical simulations in the estimates of permafrost heat storage adds uncertainties to the results. As discussed in the manuscript, ERA-Interim is now superseded by the ERA5 reanalysis, and this new reanalysis should be used in a future iteration of this work. Regarding the Mk3L paleosimulation, new models contributing to the PMIP4 project may be more suitable. In any case, please note that the Mk3L simulation is used only to initialize the permafrost model, as finding an equilibrium state for ground ice under preindustrial conditions requires several centuries, and then a transitional period between preindustrial conditions and conditions at the starting date of ERA-Interim (1979 CE) should be provided. Also note that the surface boundary conditions for $\sim 60\%$ of the period of interest are obtained from the observation-based ERA-Interim reanalysis. Therefore, the effect of using a more advanced paleosimulation should be small.*

*The largest uncertainties regarding the permafrost heat uptake are expected to be related to the effect of snow on the ground thermal regime and the distribution of ground ice. These first-order effects were addressed by our parameter ensemble simulations. Please refer to* nitzbon2022permafrostheatstora *for an extended discussion of the uncertainties and limitations of the permafrost heat uptake.*

(4). Uncertainty estimates for inland waters. Is the ensemble spread used to estimate the uncertainty of heat storage in inland waters? If so, it is fundamentally different from the other two components, i.e. the assumption underlying this method is: model difference (whatever caused the difference) can fully represent the uncertainty. Such assumption is likely wrong as there are always common model biases. And such assumption is clearly different from the assumption for your permafrost thawing and ground heating uncertainty estimate, so they can not be simply added up, simply physically meaningless.

*In our analysis, we explore and analyse each component separately, considering both spatial and temporal variability. Later, a common estimate for the entire continental system is derived. As pointed by the reviewer, using standard error propagation methods to derive the total uncertainty*

*in continental heat storage is excessively simplistic and omits critical differences in the methodology used to derive the heat storage within each subsystem. In the new version of the manuscript, we still provide with an uncertainty estimate for the continental heat storage but explaining the limitations of each method and the differences among the uncertainty estimated for each individual component. We also keep the uncertainty estimates for each individual component; thus the reader can reach their own conclusions about the trustworthiness of the reported uncertainties, and compute its own estimates.*

(5). How the final estimate of land heat storage uncertainty been derived? Are you assuming independency of the three components? Are they independent?

*The final continental heat storage series results from adding the global estimates for ground heat storage, permafrost heat storage, and inland waters heat storage using standard error propagation methods (lines 258-262 in the original manuscript). Here we should inform the reviewer of an error in the processing of the uncertainty estimates that lead to and underestimation of the total uncertainty in continental heat storage. Please, check the new version of the manuscript for an updated estimate.*

*Regarding the independence of the estimates, the three estimates are considered independent, as ground heat storage estimates do not include the heat uptake by permafrost thawing, and no ground heat storage nor permafrost thawing is possible in lakes or reservoirs. Nevertheless, the uncertainty estimate for the total continental heat storage is probably not robust, thus we have added a paragraph discussing the limitations in our estimate (see also our answer to the previous comment).*

(6). Line 219: Please explain why "this large interannual variability is explained by the smaller surface of global lakes and reservoirs in comparison with the global land and permafrost areas"?

*What we wanted to indicate here is that since inland water bodies cover a surface that is two orders of magnitude smaller than the land surface, and one order of magnitude smaller than the total permafrost area, we can expect a larger inter-annual variability in the estimated inland waters heat flux than in the other two components. We have rewritten this sentence in the new version of the manuscript.*

(7). Line 257. The total land heat storage is 23.9+/-0.4 ZJ. The error range is too small to believe. Look at Fig. 1a, there are only several places with observations, and the spatial variability is large (that means you need more data to resolve these variability), so I don't think the uncertainty can be so small. The

uncertainty estimate should be better documented in this study, and any revision should be carefully assessed and validated.

*Please, check our answer to the second comment.* cuesta-valero2022bti *explains in detail the reason for this new smaller uncertainty in global estimates of ground heat flux, and it also includes a comprehensive comparison with previous techniques to retrieve uncertainty from inversions of subsurface temperature profiles. In a nutshell, previous estimates converge to the new uncertainty results when individual inversions from subsurface profiles are aggregated correctly.*

To proceed (avoid rejection of this paper), I recommend the authors not putting the the estimates for the three estimates toghether, just presenting them separately, making a point that permafrost and lakes might be important in EEI, which is the best the authors' can do.. I disagree to put them together because some are model-based estimates, and the uncertainty etsimates are apparant very weak.

*We disagree with the reviewer. We are already providing with the individual estimates for each component of the continental heat storage, analysing their temporal and spatial variability. Nevertheless, we recognize that we underestimated the differences between the sources of uncertainty considered in each continental subsystem, and we have included a discussion about the different uncertainties in each subsystem. Thereby, the readers of the manuscript have access to the individual estimates for ground, permafrost, and inland waters heat storage, information about how to interpret these estimates, the result of applying common error propagation methods, and a warning about the limitations in this uncertainty analysis.*

*Regarding the combination of measurements and models, we refer the reviewer to our answer to the first comment: combination of observation-based results with reanalysis and modelling estimates is a common practice in the climate community when observations for a relevant variable or component of the Earth system are not available. We consider that not using reanalysis or simulations to try to better understand the behaviour of the climate system is a mistake. Nevertheless, we agree that these lines of evidence cannot replace observations, and that they are very different between each other. Therefore, we clearly identify the source of data for our estimates, we clearly indicate in our manuscript that reanalyses and model simulations include additional uncertainties not present in subsurface temperature profiles, and we indicate that observations should be incorporated into the analysis where and when possible. We have also added a paragraph to the Conclusions section clearly indicating that more observation based data should be included in the new version of this analysis.*

*Unfortunately, it is not within our immediate reach to fill the observational gaps appearing in this*

*analysis, but we can still use other sources of information for mitigating those gaps as much as possible.*

---

## Author Comment (AC3)

*Dear Reviewer,*

*We thank you for your thorough and constructive feedback. This file provides a complete documentation of the changes made in response to each of your comments. Reviewer's comments are shown in normal text, author responses are shown in bold, italic, blue text.*

**Reviewer 3**

In this manuscript the authors evaluate the continental heat uptake since 1960. It is an update of their previous work published in von Schuckmann et al. 2020. Compared to von Schuckmann et al. they made 3 changes in their estimate: 1) they changed their method to estimate the ground heat uptake from subsurface temperature profiles 2) they added an estimate of the permafrost heat uptake with a permafrost model and 3) they added an estimate of the lake, reservoir and river heat uptake with a global lake model forced by historical simulations of Earth System Models (ESM). The authors find a total continental heat storage of $23.9 \pm 0.4$ ZJ since 1960 which is consistent with von Schuckmann et al. estimate of 24 ZJ since 1960. But this consistency is by chance. Indeed, the authors actually find a ground heat uptake that is significantly smaller than von Schuckmann et al. 2020 by 12mW.m-2 and the difference in ground heat uptake is compensated by the addition of the permafrost heat uptake and the inland water heat uptake estimates.

The manuscript is clear and well written. It deals with an important question which is the distribution among reservoirs of the excess of heat gained by the climate system in response to greenhouse gases emissions. The distribution of heat among the Earth reservoirs at interannual and longer time scales is driven by the different heat capacities of the different reservoirs. The heat capacities of the reservoirs show very marginal changes with climate change thus the distribution of heat in the climate system is the same over time as it warms. It is important to estimate the distribution of heat among the Earth reservoirs to determine where the heat is actually located and what are the places of the world that are the most impacted by global warming. This is also a good indicator of current global warming and its distribution. As such, it is useful to derive the land heat uptake in order to raise public awareness. The work in this manuscript helps in this objective. In particular I find interesting the tentative estimate of the permafrost heat uptake. Permafrost heat uptake is an important indicator of the changes in a key place for the future of the climate system. It could definitely be an interesting indicator to raise public awareness.

Concerning the results of this paper, I find they are disappointing for three main reasons

First the authors find a ground heat uptake that is not consistent with their previous estimate in von Shuckmann et al. 2020 although they have used the same subsurface temperature profiles and the same inversion method. The significant difference between their previous estimate and the current estimate comes from the aggregation technique. But the confidence in the aggregation technique is not evaluated in the manuscript. So, we don't know why the estimate of ground heat uptake is so sensitive to the aggregation technique and what should be done to tame down this high sensitivity. We don't know either which aggregation technique should be trusted and thus which estimate of the ground heat uptake should be trusted: the one that is propsoed in this manuscript or the previous one from von Shuckmann et al. 2020? More analysis are needed here to determine the confidence in the ground heat uptake estimate and explain the causes for the differences among the different estimates

*Please, note that the bootstrapping aggregation method has been extensively described and analysed in Cuesta-Valero et al. (2022), including a comparison with the aggregation method used in von Schuckmann et al. (2020). This reference was a preprint at the time of submitting our manuscript, but now it has been peer-reviewed and it is fully published.*

*Of course, we consider that the ground heat storage estimate in this paper is better than the one published in von Schuckmann et al. (2020) because of the reasons outlined here and in Cuesta-Valero et al. (2022).*

Second, the authors estimate the inland water heat uptake from models only. They do not use any observations or reanalysis (even the forcing of the global lake model is coming from ESMs). If the objective of continental heat uptake estimates is to "inform about future warming and climate change as well as to understand the future consequences for society and ecosystems associated to continental heat gains» as the authors claim, then it does not make sense. Climate model projections are not informed by their own simulations of the historical period. They are informed by comparison against independent observations retrieved from the real world. So, to support their objectives the authors should provide an estimate of the inland water heat uptake that is derived from observations in a way or another (using forcing from reanalysis for example?)

*We agree with this point. Nevertheless, we have no alternative right now. ISIMIP currently is the only source of multi-lake-model estimates available globally. The reason for using forcing from climate models instead of reanalysis is that in ISIMIP2 such reanalysis-driven runs simply have not been conducted. Moreover, if they would have been conducted, they would have stopped in the recent past (2015 in ISIMIP2a and 2019 in ISIMIP3a), thus even then we would be missing the most recent years. Furthermore, while the climate models do not capture natural variability, they have been bias-adjusted, and they are designed to capture long term meteorological trends (that*

*drive inland changes in water conditions).*

*As discussed on the manuscript, this is an aspect of the study to be improved in the future, but we think there is no immediate solution.*

Third I find that the uncertainty estimates are in general largely overlooked over the whole paper. In the case of the ground heat uptake, we are left at the end of the paper with a new estimate of the ground heat uptake with a very low uncertainty range ($\pm$0.8mW.m-2). This small uncertainty range only accounts for errors in the thermal diffusivity, errors in the thermal conductivity and errors in the reference profile (through the bootstrap approach). But it does not account for any sources of systematic uncertainty such as the poor and inhomogeneous distribution of the subsurface temperature profiles. Given the high sensitivity of the ground heat uptake estimate to the aggregation technique, the poor distribution of subsurface data is certainly the dominant factor of uncertainty here. Thus the very small uncertainty range of $\pm$0.8mW.m-2 is dubious. In the case of the permafrost heat uptake the uncertainty range does not account for many sources of systematic uncertainties as well. In particular the estimate is done with a unique permafrost model. Permafrost models show very large differences. At least, the use of another (or several) model would give insights on the level of this potentially large source of systematic uncertainties. In the case of the inland water heat uptake there is simply no information on how the uncertainty is derived.

*Previous works have concluded that the distribution of subsurface temperature profiles is enough to provide with hemispheric and global averages considering profiles (Beltrami et al., 2004; Pollack et al., 2004), pseudo-proxy experiments using climate simulations (e.g., Beltrami et al., 2006; González-Rouco et al., 2006; García-García et al., 2016; Melo-Aguilar et al., 2020), and CRU TS data (Cuesta-Valero et al., 2021). Therefore, we consider the incomplete distribution of profiles to be a minor source of uncertainty. Regarding the width of the uncertainty range for ground heat flux, please see again Cuesta-Valero et al. (2022) and our answer to the first comment. We have added a sentence explaining why we think the spatial distribution of subsurface temperature profiles is not a significant limitation in the new version of the paper.*

*In the case of permafrost, uncertainties arise from the different stratigraphies and snow cover characteristics considered, trying to address all the poorly known, or directly unknown, configurations of ground ice, water and snow in the Arctic. Indeed, considering different permafrost models would allow us to include the uncertainty due to the different model physics. Nevertheless, our results already present a wide uncertainty range that is unlikely to be exceeded using further models. Please see Sections 2.2 and 5 for more details.*

*Uncertainties in inland waters heat storage are estimated from the multimodel ensemble analysed here. Concretely, the best estimate consists in the average of all sixteen lake model simulations, with the uncertainty range being defined by the standard deviation. Since the ensemble consist of four different global lake models, each driven by four bias-corrected GCM simulations, providing the climate forcing, the estimates include both the uncertainties related to structural differences in the lake models, and to the different climate trajectories simulated by the driving GCMs. We have included a line in Section 2.3 explaining this point in the new version of the text.*

*Standard error propagation methods are applied to drive the uncertainty of the continental heat storage from the different uncertainty estimates of the three components. Nevertheless, we recognize that this is not a robust approach given the differences in the factors contributing to the uncertainties in the three components of the continental heat storage. We have included a discussion about this in the text.*

For these reasons I think the paper is not ready for publication as it is. I think it needs a substantial amount of work to answer the important points I raised before.

*We respectfully disagree with the reviewer. We are well aware of the limitations of the analysis, we detail them in the manuscript, and provide a plan for reducing current limitations in future iterations of the analysis.*

*Arguably, the main limitation of the analysis is the extensive use of models to provide estimates for subsystems without adequate in-situ measurements. Unfortunately, the lack of adequate observations is not going to disappear in the short term, but we think that model simulations can be used in the meantime as best-guesses. This is not the ideal situation but a sort-term solution to inform the scientific community about the thermal state of the surface and subsurface of the continental landmasses, as well as to estimate the Earth heat inventory as precisely as possible.*

*We have added a new paragraph in the Conclusions section clearly stating that the current estimates of continental heat storage should be improved in future iterations of this analysis, and we have included a discussion about the different uncertainties in the estimates of continental heat storage. We think that these two additions enhance the clarity of the current analysis and provide with a clear path for future iterations of these project.*

I add below a list of additional comments

L46: what do you mean by "consistently"

*We mean that the land term of the Earth heat inventory has been the second largest term after the ocean in previous analyses. We have changed this sentence to improve the clarity.*

L53:"high latent heat of fusion": high compared to what?

*We refer here to the high energy required to melt a certain amount of water in comparison to the energy required to warm it. We have clarified this point in the new version of the manuscript.*

L95: the Xibalba logs are poorly and in-homogeneously distributed. Have you estimated the biases that could be caused by this in-homogeneous distribution? This is probably a leading source of uncertainty. You should at least estimate the order of magnitude of this source of systematic uncertainty and acknowledge it in the paper.

*Please, check our answer to the third comment above. Several works have shown before that the global distribution of subsurface temperature profiles is complete enough to represent the long-term evolution of surface conditions at the global scale.*

L102: the reference period for the calculation of the quasi equilibrium is precisely during the little ice age when land heat uptake was probably negative. This is potentially an issue for the inversion as it may bias high the anomalies with respect to the quasi equilibrium (since the quasi equilibrium you chose was a cold transient response to the little ice age rather than an equilibrium). Have you analyzed this possibility ? Do you have an idea of the potential error induced by the fact that the reference period is during the little ice age rather than during an equilibrated period?

*Indeed, the period of reference for borehole inversions include part of the Little Ice Age (LIA, 1300-1850). Nevertheless, we consider the potential effect of LIA in our estimates of ground heat storage to be small. As indicated in the manuscript, we estimate heat storage as accumulated heat flux since 1960, that is, in our estimates flux is 0 in 1960, thus the LIA signature in the estimated heat storage should be irrelevant. There should be an effect on the flux histories retrieved from logs containing LIA signatures, but since LIA was not a spatially homogeneous process, not all profiles will include this signature (e.g., Beltrami et al., 2003).*

*Furthermore, the number of eigenvalues retained in the solution also limits the presence of the LIA signature in the inversions (Melo-Aguilar et al., 2020). Since we used the two highest eigenvalues in our inversions, the effect of LIA in the inversions will probably be attenuated. Therefore, we conclude that LIA has not a relevant role in our results.*

*Another issue with a possible change of depths to estimate the quasi-equilibrium profiles is the fact*

*that the number of logs quickly decreases with depth requirements. That is, a change in the depth range to provide estimates relative to a period before the LIA would result in a markedly lower number of profiles. And a change towards shallower depths risks including the transient signal in ground temperatures due to the beginning of the industrialization, biasing the determination of the quasi-equilibrium profile.*

*For these reasons, we consider the effect of LIA to be small, and the current depth range to estimate the quasi-equilibrium profiles of 200-300 m as the one that avoids the transient signal of the Industrial Revolution on the profiles while maximizing the number of logs contributing to the global ground heat storage.*

L130: same remark as for L95: the bootstrap approach quantifies the uncertainty due to errors in the thermal diffusivity, errors in the thermal conductivity and errors in the reference profile. But what about the systematic errors coming from the in-homogeneous and poor distribution of profiles? This source of uncertainty probably dominates over the others. Can you elaborate on this? Provide a first estimate of this systematic error?

*Please, check our answers to previous comments.*

L145: the permafrost heat storage is derived from a model. But to which extent can we trust this model to represent the actual Permafrost? You do not provide any information on the validation of the model against observations. What confidence do we have in such a model?

*The model has been compared to ground surface temperature measurements at 82 different permafrost stations from the Global Terrestrial Network for Permafrost (GTN-P) covering the period 2007-2016 (Langer et al., 2022). The root mean squared error between the simulation and the measured temperatures is reported to be 2.2 K, with a warm bias of 0.6 K, a performance comparable or better than other model analyses. For a detailed evaluation of the model, please check the indicated reference. We have also included a couple of lines with these results in the new version of the text.*

L160: you are using a unique permafrost model. What about comparing against other independent models to get insight on the amplitude of potential sources of systematic uncertainty related to your model?

*Please, check our answers to the previous comments. We know that this is an important issue that shall be mitigated in the next iteration of this analysis.*

L172: why not using a forcing from reanalysis rather than ESM? This would be much closer to the real world. You claim further that the heat uptake estimate is important to inform projections of the future climate. If so, you need to get observational estimates of the heat uptake rather than model estimates. I don't understand the rationale here to use ESM forcing rather than reanalyses forcing

*The rationale here is as follows: we want to have an ensemble of multiple lake models to remove the effect of structural lake model bias. The ensemble is produced by a community of lake modellers usually after years of discussion and planning, thus it takes a substantial amount of time to implement design changes in ISIMIP protocols. The point raised by the reviewer is a good argument for pushing for reanalysis-driven simulations in the next iteration of ISIMIP, ISIMIP3a. But not all modellers would be interested in this type of simulations, and in any case ISIMIP3a will cover until 2019, so the produced experiments would not cover the recent past.*

*Unfortunately, there is no immediate alternative, as ISIMIP currently is the only source of multi-lake-model estimates available globally.*

L199: How do you account for river depth?

*River depth is dealt with within both global hydrological models (WaterGAP2 and MATSIRO) in their simulation of the total water stored in rivers. In WaterGAP2 for example, river water storage of a grid cell is calculated based on the hydraulic radius, which is based on the actual discharge and empirical relationships between river depth and width at bankfull conditions (Müller Schmied et al., 2014).*

L199: how do you compute the uncertainty of your inland water heat uptake estimate?

*Please, check out answer to previous comments regarding this point.*

L204: the new uncertainty range is one order of magnitude smaller!!! This is huge! Especially for uncertainty. How do you explain that?

*In short, previous works considering several subsurface temperature profiles overestimated the 95 % confidence interval, as their aggregation methods were not correct from a statistical point of view. For a detailed analysis and a comparison between the new bootstrapping aggregation technique and previous methods, check Cuesta-Valero et al. (2022). We have reinforced this explanation in the new version of the manuscript too.*

L204: the very small uncertainty of the present study is such that your result is inconsistent with your

previous estimate in von Shuckmann et al. How do you explain that? The inconsistency between both results means that one or the other or both estimates are wrong!! Which one is wrong then? The present study estimate or your previous study estimate? The paragraph L202 to l212 recall the method used in von Shuckmann et al. 2020 and the method used here to aggregate the data. But it is inconclusive on which aggregation method should be trusted. Since the two methods yield inconsistent results, we need to understand where the problem is, which number should be trusted and why we should trust it rather than the other.

*Please, check our answer to the previous comments. The new estimate is the better one because of the reasons outlined above.*

L215: I find dubious that the different inversion technique and the different number of vertical profiles are enough to explain a change of the ground heat uptake by a factor 2 between Beltarmi 2002 and this study. Either there is a misunderstanding of the real causes for the difference between both estimates or it means that land heat uptake is highly sensitive to the number of vertical profiles. It brings me back to my previous question: is there an important bias due to the poor and in-homogeneous sampling of the vertical profiles. A good test would be to take the same profiles as Beltrami 2002 and re estimate the land heat uptake with the inversion developed here and check whether you find the same result

*Redoing the analysis of (Beltrami et al., 2002) (BE02 hereinafter) is not completely possible, as Xibalbá profiles are truncated at $300$ m, and profiles in BE02 were used with full depth. Nevertheless, we can try to mimic the methodology of this analysis within certain limitations. There are 910 Xibalbá profiles measured before 2002, in comparison to the 616 logs reported in BE02. Inverting these 910 selected profiles with a step change of 50 years for the solutions (as in BE02) leads to a ground heat flux of $58$ mW m$^{-2}$ for the period 1950-2000, which is more similar to the $\sim 40$ mW m$^{-2}$ in BE02 than to the $\sim 85$ mW m$^{-2}$ reported for 1960-2020 on the text. The remaining difference can be explained by I) the almost 300 additional logs in comparison with BE02, and II) the fact that logs were not truncated to the same depth in BE02, thus each log had a different reference for estimating the quasi-equilibrium temperature profile. These results are consistent with what we indicated in the previous version of the paper.*

L262-264 your new result agrees with your old result but for wrong reasons!! It is because you were biased in the ground heat estimate and here the bias is compensated by the new reservoirs you are adding in (permafrost and lakes). The right conclusion is that you find a ground heat uptake that is significantly different from the previous one. You should acknowledge that clearly and explain why. Can you elaborate on that?

*Indeed, we obtain a different ground heat storage than in von Schuckmann et al. (2020), as we have pointed out in the manuscript. Also, please check the detailed analysis in the reference Cuesta-Valero et al. (2022) indicated on the text. We have changed the text in order to improve the clarity of this point.*

L270 paragraph 4: I don't understand the point of this paragraph. Indeed we know that land heat uptake has numerous impacts on society and ecosystem. But it does not mean we need to estimate the land heat uptake to anticipate those impacts. In practice impacts on society and ecosystem are not derived from estimates of the land heat uptake. They are rather estimated from the output of climate models which use as input $CO_2$ concentrations and which tune their model against surface temperature and global EEI at TOA. So, in which way estimating land heat uptake will help to improve climate models and anticipate impacts on society or ecosystems. We should rather focus on improving the land surface models that are embedded in climate models, shouldn't we?

*In the mentioned paragraph, we did not mean that estimates of ground heat storage are required to quantify or to anticipate the impacts, but that climate simulations suggest that ground heat storage is going to keep increasing in the near future, even in low emission scenarios, as well as the impacts associated to heat storage. We have rewritten the conflicting sentences to make that clear.*

*Also, quantifying ground heat storage helps to identify shortcomings in climate models, particularly in land surface model components. For example, previous works have shown that climate simulations underestimate ground heat storage because their land surface models are too shallow, and therefore the represented volume of the continental subsurface is insufficient to store the right amount of heat. This bias in the thermal state of the subsurface also affects the simulated subsurface temperatures, as well as the represented amount of permafrost in the model. There is a discussion about this point in the Conclusions section of the manuscript.*

The only interest I see in estimating land heat uptake or permafrost heat uptake is to derive indicators for public awareness. Is that what you want to do? If so, you should state it clearly

*We respectfully disagree with the reviewer about this point. We consider that estimating the heat stored in all climate subsystems is also important to quantify the evolution of the Earth energy imbalance at top-of-the-atmosphere, as explained in von Schuckmann et al. (2020) or Chapter 7 of the sixth assessment report of the IPCC (Forster et al., 2021). Of course, the ocean accounts for $\sim 90\%$ of the total heat storage but ignoring the rest of the subsystems imply to add a bias of $\sim 10\%$ to the estimate, a bias that can be avoided.*

L322: I have the same remark: I don't see how the magnitude of land heat uptake inform on future warming and climate change. Futur warming and climate change are given by climate models and climate model just don't work with land heat uptake. So please elaborate to explain what you mean here

*Indeed, a better wording was possible for this sentence. We have rewritten it in the new version of the manuscript.*

L325: An interest I see in estimating land heat uptake is to derive an observational benchmark against which climate model could be validated. But in this case you would need to derive observation only estimates of land heat uptake. That would be probably more suitable to the objective of informing projections of future warming

*We completely agree with the reviewer. This is the beginning of a collaboration to provide with such estimate, but while we investigate the way of obtaining such observation-based estimates, we are forced to relay in reanalysis and models. As there seems to be a confusion regarding the long-term goal of this collaboration, we have provided with a new final paragraph clearly stating our plans for the future.*

**References**

Beltrami, H., Smerdon, J. E., Pollack, H. N., and Huang, S. (2002). Continental heat gain in the global climate system. *Geophysical Research Letters*, **29**(8), 8–1–8–3. DOI: 10.1029/2001GL014310.

Beltrami, H., Gosselin, C., and Mareschal, J. C. (2003). Ground surface temperatures in Canada: Spatial and temporal variability. *Geophysical Research Letters*, **30**(10). DOI: https://doi.org/10.1029/2003GL017144.

Beltrami, H. and Bourlon, E. (2004). Ground warming patterns in the Northern Hemisphere during the last five centuries. *Earth and Planetary Science Letters*, **227**(3–4), 169 –177. DOI: http://dx.doi.org/10.1016/j.epsl.2004.09.014.

Beltrami, H., Bourlon, E., Kellman, L., and González-Rouco, J. F. (2006). Spatial patterns of ground heat gain in the Northern Hemisphere. *Geophysical Research Letters*, **33**(6). n/a–n/a. DOI: 10.1029/2006GL025676.

Cuesta-Valero, F. J., García-García, A., Beltrami, H., González-Rouco, J. F., and García-Bustamante, E. (2021). Long-term global ground heat flux and continental heat storage from geothermal data. *Climate of the Past*, **17**(1), 451–468. DOI: 10.5194/cp-17-451-2021.

Cuesta-Valero, F. J., Beltrami, H., Gruber, S., García-García, A., and González-Rouco, J. F. (2022). A new bootstrap technique to quantify uncertainty in estimates of ground surface temperature and ground heat flux histories from geothermal data. *Geoscientific Model Development*, **15**(20), 7913–7932. DOI: 10.5194/gmd-15-7913-2022.

Forster, P., Storelvmo, T., Armour, K., Collins, W., Dufresne, J.-L., Frame, D., Lunt, D., Mauritsen, T., Palmer, M., Watanabe, M., Wild, M., and Zhang, H. (2021). "The Earth's Energy Budget, Climate Feedbacks, and Climate Sensitivity". In: *Climate Change 2021: The Physical Science Basis. Contribution of Working Group I to the Sixth Assessment Report of the Intergovernmental Panel on Climate Change*. Ed. by V. Masson-Delmotte, P. Zhai, A. Pirani, S. Connors, C. Péan, S. Berger, N. Caud, Y. Chen, L. Goldfarb, M. Gomis, M. Huang, K. Leitzell, E. Lonnoy, J. Matthews, T. Maycock, T. Waterfield, O. Yelekçi, R. Yu, and B. Zhou. Cambridge, United Kingdom and New York, NY, USA: Cambridge University Press, 923–1054. DOI: 10.1017/9781009157896.009.

García-García, A., Cuesta-Valero, F. J., Beltrami, H., and Smerdon, J. E. (2016). Simulation of air and ground temperatures in PMIP3/CMIP5 last millennium simulations: implications for climate reconstructions from borehole temperature profiles. *Environmental Research Letters*, **11**(4), 044022. DOI: 10.1088/1748-9326/11/4/044022.

González-Rouco, J. F., Beltrami, H., Zorita, E., and von Storch, H. (2006). Simulation and inversion of borehole temperature profiles in surrogate climates: Spatial distribution and surface coupling. *Geophysical Research Letters*, **33**(1). n/a–n/a. DOI: 10.1029/2005GL024693.

Langer, M., Nitzbon, J., Groenke, B., Assmann, L.-M., Schneider von Deimling, T., Stuenzi, S. M., and Westermann, S. (2022). The evolution of Arctic permafrost over the last three centuries. *EGUsphere*, **2022**, 1–27. DOI: 10.5194/egusphere-2022-473.

Melo-Aguilar, C., González-Rouco, J. F., García-Bustamante, E., Steinert, N., Jungclaus, J. H., Navarro, J., and Roldán-Gómez, P. J. (2020). Methodological and physical biases in global to subcontinental borehole temperature reconstructions: an assessment from a pseudo-proxy perspective. *Climate of the Past*, **16**(2), 453–474. D O I: 10.5194/cp-16-453-2020.

Müller Schmied, H., Eisner, S., Franz, D., Wattenbach, M., Portmann, F. T., Flörke, M., and Döll, P. (2014). Sensitivity of simulated global-scale freshwater fluxes and storages to input data, hydrological model structure, human water use and calibration. *Hydrology and Earth System Sciences*, **18**(9), 3511–3538. D O I: 10.5194/hess-18-3511-2014.

Pollack, H. N. and Smerdon, J. E. (2004). Borehole climate reconstructions: Spatial structure and hemispheric averages. *Journal of Geophysical Research: Atmospheres*, **109**(D11). n/a–n/a. D O I: 10.1029/2003JD004163.

Von Schuckmann, K., Cheng, L., Palmer, M. D., Hansen, J., Tassone, C., Aich, V., Adusumilli, S., Beltrami, H., Boyer, T., Cuesta-Valero, F. J., Desbruyères, D., Domingues, C., García-García, A., Gentine, P., Gilson, J., Gorfer, M., Haimberger, L., Ishii, M., Johnson, G. C., Killick, R., King, B. A., Kirchengast, G., Kolodziejczyk, N., Lyman, J., Marzeion, B., Mayer, M., Monier, M., Monselesan, D. P., Purkey, S., Roemmich, D., Schweiger, A., Seneviratne, S. I., Shepherd, A., Slater, D. A., Steiner, A. K., Straneo, F., Timmermans, M.-L., and Wijffels, S. E. (2020). Heat stored in the Earth system: where does the energy go? *Earth System Science Data*, **12**(3), 2013–2041. D O I: 10.5194/essd-12-2013-2020.

---

## Author Comment (AC4)

*Dear Reviewer,*

*We thank you for your thorough and constructive feedback. This file provides a complete documentation of the changes made in response to each of your comments. Reviewer's comments are shown in normal text, author responses are shown in bold, italic, blue text.*

**Reviewer 1**

General Comments

The manuscript submitted by Cuesta-Valero et al. considers continental heat storage and determines the contribution from three components. The analysis is important as it contributes to better understanding of the overall global heat balance by ensuring that all components are accounted for in the calculation of continental heat storage. The subject area is therefore appropriate for publication in ESD and would be of interest to its readers. The MS is also relevant to better estimates of the impact of climate change on the landmass. The MS has clear objectives and is generally well written with results and interpretations presented clearly. I don't have any major concerns with the MS but I do have a number of comments that should be considered prior to acceptance for publication.

One of the key things that is done in the paper is the calculation of the heat in the ground that is utilized for phase change (latent heat) as ice in permafrost melts. However, the way the paper is written the authors seem to consider this separate from the subsurface (or ground) heat storage, which I found odd. Permafrost is a component of the ground (essentially a thermal condition of the ground) in cold environments so both the heat used to raise its temperature or for phase change when it thaws are components of the heat that is stored in the ground. It would seem that this is more an issue of the method that has been traditionally utilized to determine ground heat storage. Analysis utilizing subsurface temperature profiles only considers conduction in the estimate of ground heat fluxes. As ground temperatures approach 0 °C in permafrost, heat is utilized for phase change of any ice in the ground rather than raising the temperature and little change in temperature over time is observed in ground temperature profiles (as discussed in Romanovksy et al. 2010; Smith et al. 2010). Lack of consideration of the latent heat effects therefore means that ground heat storage determined considering only conduction would be underestimated. It would make more sense for the authors to say that they are refining the estimates of ground heat storage by addressing a limitation of the method traditionally used by considering the latent heat utilized for phase change in the estimates.

*The reviewer is right that permafrost is just perennially frozen ground and that our permafrost heat storage estimate is essentially the change in latent heat storage. Furthermore, available methods to estimate ground heat storage from subsurface temperature profiles cannot include latent heat flux used to thaw permafrost, as indicated by the reviewer.*

*In our analysis, we use a model and a series of assumptions about the stratigraphies of the Arctic subsurface in order to estimate the latent heat used in permafrost thawing, in order to complement the observation-based method used to derive sensible ground heat storage. That is, we separate the sensible and latent heat fluxes, mainly due to methodological limitations. Therefore, we believe that it is better if we maintain both estimates of heat storage as separate entities in order to improve the clarity of the manuscript.*

*We have added a couple of lines in the new version of the manuscript to make clear the division into sensible and latent heat fluxes.*

The authors do not mention the role of other modes of heat flux in the ground such as convection. Heat transfer associated with water movement (advection) such as infiltration of precipitation and snow melt or subsurface water flow may also influence the ground thermal regime (see for eg. Douglas et al. 2020; Neumann et al. 2019; Phillips et al. 2016; review of Smith et al. 2022b also discusses this). As permafrost thawing occurs, subsurface water flow becomes more important. Is lack of consideration of this mechanism of heat flow also a limitation of the method used to determine ground heat storage?

*Indeed, our approach to derive estimates of permafrost heat storage is not able to include an active hydrology in the subsurface. We have noted this fact as a limitation in the manuscript.*

*Regarding advection in subsurface temperature profiles, the diameter of the drilling holes is usually small enough to prevent air advection. Water advection is still possible, which may introduce a non-climatic signal in the profiles. Nevertheless, all logs were screened by eye, and those including signals that cannot be explained by climate alone were removed (see the details in Cuesta-Valero et al., 2021).*

*We have added a couple of lines in the new version of the manuscript clarifying this point.*

I have a number of additional comments (see below) for the authors' consideration in preparing the revised manuscript. These comments identify where further clarification or information may be required. Suggestions for editorial revisions have also been provided.

Specific comments (keyed to line number)

L31 – See comment above – permafrost is the ground (earth material) so its thaw is a component of subsurface heat storage.

*We have already addressed this comment above.*

L32 – Suggested revision: " The ground accounts for  90

*Done.*

L41 – What is included in "cryosphere"? Permafrost is a component of the cryosphere but it is treated separately in this paper.

*In this context, the term cryosphere refers to glaciers and ice caps. We have indicated this in the new version of the manuscript.*

L53 – Permafrost includes soil and rock. Since there can be water within rock, phase change can also occur in frozen rock (even if the amount is small compared to soils).

*We have changed the text to reflect this point.*

L55 – replace "underline" with "underlie"

*Done.*

L55 – Note Obu (2021) determines the equilibrium permafrost distribution so it does not consider permafrost that formed under a colder climate and still persists today. For example, permafrost in peatlands in the southern portion of the permafrost regions formed under colder conditions and is preserved due to the insulating properties of peat. Also, permafrost can be quite thick in the Arctic and it can take a century or more to completely thaw so that relict permafrost continues to exist as climate warms.

*This is correct. We wanted to give an idea of the total area underlain by permafrost. Please note that the reported warming for permafrost after the Obu (2021) reference corresponds to recent times.*

L56 – It is important to note that these are average values of warming based on several sites (I believe Biskaborn 2019 gives a range).

*We have added the uncertainty ranges to the new version of the manuscript.*

L59 – Misleading/incorrect statement. These simulations only consider the upper 2-3m of permafrost

rather than its total vertical extent, which may be 10s to 100s m. These values therefore do not refer to complete loss of permafrost from this area (i.e. refer to thaw being more than 2-3m over this area).

*CMIP simulations from Koven et al. (2013), Slater et al. (2013) and Burke et al. (2020) consider only shallow permafrost. Nevertheless, the LSMs considered in Hermoso de Mendoza et al. (2020) and in Steinert et al. (2021) consider soil columns with hundreds of meters of depth. However, we agree with the reviewer that the range of change in global permafrost extension refers to shallow permafrost, thus we have changed this in the new version of the paper.*

L61 – Permafrost is frozen ground so permafrost heat uptake is ground heat uptake. Until it thaws, the heat storage would be accounted for by the methods (inversion of temperature profiles) utilized to determine ground heat storage.

*We completely agree with the reviewer. Because of this, we refer only to the change in the area of permafrost in the previous line, and not to the change in permafrost temperature.*

L66 – What is meant by recent times? It would be clearer to give the time period over which this reduction occurred.

*We meant the last three decades. We have included this period on the text.*

L67 – suggested revision: ' ....going to continue throughout the 21st century...:

*Done.*

L79 – should this be "deep subsurface temperature profiles"

*Done.*

L87 – replace "in" with "of"

*Done.*

L89 – revise to "slope of this regression line" (or best-fit line)

*Done.*

L99-100 – If the time for temperature changes at the surface to reach a given depth depends on the thermal properties, how does truncating to the same depth yield the same temporal reference if thermal properties are variable?

*The reviewer is right, the time required for a surface perturbation to reach a certain depth depends on thermal properties. What we are assuming to provide the temporal reference indicated in line 101 is an homogeneous subsurface with a thermal diffusivity of $1.0 \times 10^{-6} \text{ m}^2 \text{ s}^{-1}$, which is a typical value for bedrock. We have modified the text to clarify this point.*

L131-134 – I may have missed something here - how are the results from point-based measurements applied to the entire area considered. In figure 2a, heat storage is shown for points that are not uniformly distributed with very large areas not represented. It isn't clear how the point-based data are extrapolated to the larger area or what other information may be utilized especially give the large areas with no data.

*We followed the methodology in Cuesta-Valero et al. (2021), consisting in obtaining the averaged heat flux from all 1079 subsurface temperature profiles, and then estimating the accumulated heat considering a global land surface of $1.34 \times 10^{14} \text{ m}^2$. That is, we consider the area of all continents excluding Antarctica and Greenland, since we have no measurements there. This is possible because previous works have shown that the current distribution of boreholes is enough to capture global changes in surface conditions (e.g., Pollack et al., 2004; García-García et al., 2016). Furthermore, Cuesta-Valero et al. (2021) showed that changing the area considered does not affect the global estimates.*

*We have changed this paragraph in the new version of the manuscript to enhance the clarity of the text.*

L136 – Isn't it more correct to say that the heat input to the subsurface is utilized to melt ground ice as permafrost temperatures approaches 0 °C?

*Indeed, that is the complete physical process: permafrost thaws once the ground temperature is near zero Celsius degrees and the heat keeps getting into the ground. We have added a couple of lines in the text to explain the entire process.*

L140 – Do you mean the surface offset which is the difference between mean annual air and ground surface temperatures and is influenced by snow cover. The thermal offset refers to the difference in temperature between the ground surface and the top of permafrost, which (if equilibrium conditions exist) depends on difference between frozen and unfrozen thermal conductivity (See for e.g. Riseborough et al. 2008).

*We fully agree with the reviewer and changed the formulation accordingly.*

L143 – What about rock – permafrost includes rock which can contain ice.

*We used the dataset by Pelletier et al. (2016) to set the soil thickness and assumed bedrock underneath. The water/ice content in the bedrock was reduced compared to the overlying soil. Both parameters (soil thickness and bedrock ice content) were varied during the ensemble simulations to address the related uncertainties. Please see Langer et al. (2022) for details.*

L179 – How is depth determined?

*The lake depth is given by the Global Lake Database v.3(Choulga et al., 2019), as indicated in line 176 of the original manuscript.*

L165-199 – Lakes can form or drain in the Arctic due to permafrost thaw. Is the change in surface water distribution due to thermokarst processes considered or is this a limitation to heat storage estimates?

*Unfortunately, the permafrost model used here cannot represent thermokarst processes nor water redistribution. We detailed those limitations in line 341 of the original manuscript. Furthermore, we did not consider dynamic (thermokarst) lake changes in the inland water heat storage which relied on a static lake distribution. We would like to note that the overall trend of thermokarst lake dynamics is very uncertain since both lake expansion and drainage happen concurrently. For the study period of the past few decades, the net lake area change is likely negligible compared to the total lake area.*

L220 (also elsewhere in paper including L223) – See earlier comments. Permafrost heat flux, if thaw is not is not occurring (this will be the case where temperature below melting point of ice in the ground) will be considered in the estimates of subsurface storage determined utilizing subsurface temperature records. It is only when thaw occurs in warmer permafrost at temperatures near 0 °C that latent heat needs to be considered in addition to conduction.

*Please, see comment about L136 above. We only consider permafrost heat flux as latent heat flux. Permafrost warming is only considered from subsurface temperature profiles. We have added a clarification in Section 2.2.*

L235 – Where around Hudson Bay? There was cooling in the eastern Arctic including northern Quebec into the 1990s – is this the reason for the lack of heat gain in this area?

*Indeed, there is a decrease in inland waters heat storage in the southwestern shore of the Hudson Bay. (Figure 2 of the original manuscript). Unfortunately, we are unable to explain this result, and we found no explanation in the literature either. We have reported this issue in the new version of the manuscript.*

L267 – Why isn't the Tibetan Plateau included given it is a fairly significant area. Permafrost in this region is generally warm so latent heat effects are important.

*Indeed, the Tibetan Plateau is an important region that should be included in the analysis. However, the simulated permafrost relied on an input dataset of soil organic carbon (Hugelius et al., 2014) which is only available for the northern permafrost region excluding the Tibetan plateau. It is planned to include the Tibetan Plateau in the next iteration of this analysis, as indicated in the manuscript.*

L275-276 – It is important to indicate here that the estimate of ground heat flux needs to consider non conductive heat flow (i.e. address limitations) to improve estimates. The MS makes progress in addressing this limitation by considering the latent heat associated with phase change as permafrost thaws.

*We think that advection is not significant for ground heat flux at the global scale. For example, Huang (2006) uses meteorological observations of surface air temperature (SAT) to derive the evolution of global ground heat flux, reaching similar results to those in (Beltrami, 2002) from subsurface temperature profiles (GST). If nonconductive processes were relevant at the global scale, these two estimates should be different, as SAT observations would not account for these additional processes. We find that this result indicates that heat transport by conduction is the leading mode of heat diffusion trough the subsurface, with the exception of permafrost soils where thawing/freezing is occurring. Furthermore, Xibalbá profiles were screened to remove logs including advection (Cuesta-Valero et al., 2021), as indicated in the manuscript.*

L280-300 – This section is OK but most of this has been well covered in other publications so nothing really new here.

*Indeed, this part of the text is based on previous publications. Our aim was to reflect the most important results related to permafrost heat storage in order to provide a picture of the impacts that permafrost thawing posses for society and ecosystems. We have added also a small comparison with other components of the cryosphere in the new version of the manuscript in order to place our estimate in the context of the global ice budget.*

L280-285 –Other implications of ground warming and permafrost thaw are impacts on landscape processes and stability, changes to surface water distribution and increase in subsurface water flow. These impacts can also have feedbacks to the ground thermal regime with further impacts on carbon feedback.

*We have added these points in Section 4 of the new version of the manuscript.*

L288-290 – This is really an issue of landscape change associated with thawing of ice-rich permafrost (such as subsidence, thaw slumps), which is abrupt or sudden, exacerbating permafrost thaw – with geomorphic change such as slumps and other slope failures the upper boundary changes as material is removed (also lateral heat flow).

*Please, see the added text to answer the previous comment.*

L293 – Do you mean "surpassing" rather than "trespassing"

*Yes, we meant "surpassing". This is now fixed on the text.*

L295-300 – Other impacts related to permafrost thaw (especially if ice-rich) include loss of bearing strength and ground settlement/subsidence with impacts on infrastructure; landscape instability including slope failures which can release sediment into water bodies with implications for water quality; impacts on integrity of contaminant containment facilities.

*Please check Section 4 in the new version of the manuscript, we have noted those points in there.*

L301-303 – more evaporation?

*Indeed, global lakes have experienced larger evaporation rates in recent decades. Nevertheless, the leading factors causing this evaporation increase seem to be ice cover reductions (Wang et al., 2018; Zhao et al., 2022). However, for low latitude lakes, evaporation could increase by the process that lake surface temperatures warm at a slower rate than the overlying air, which leaves more energy from long-wave radiation available for lake evaporation (Wang et al., 2018). We have indicated this in the new version of the text.*

L325-335 – There are several recent ground temperature records in the permafrost regions (some results included in Smith et al. 2022b, Noetzli et al. 2022, Biskaborn et al. 2019 and other papers). These are generally at shallower depths (usually upper 20 m) than would be utilized for the inversion of ground temperature profiles that is utilized in the MS. However, these provide information at depths where latent heat effects are important.

*We are aware of those ground temperature measurements, and we are planing to include them in a future iteration of this analysis. We have added some lines in the new version of the manuscript to clarify this point.*

L337 – This is not a new observation and the lack of ground ice information has been identified as a limitation in permafrost modelling in other papers (e.g. Smith et al. 2022b; O'Neill et al. 2020).

*Yes, this is not a new result. However, this is an important limitation affecting our results, thus we think that an explanation must be included in the text for completeness and transparency.*

L347 – With respect to latent heat effects related to permafrost thaw, including the Tibetan Plateau is probably more important than permafrost zones of Antarctica given the rather dry conditions and the geology.

*Correct, and because of that we plan to include the Tibetan Plateau as soon as computational and financial resources are available, moving towards achieving global coverage in a later iteration.*

L358-359 – While the deeper subsurface is an improvement, the LSMs still have limitations with respect to representation of subsurface conditions including ground ice distribution.

*Indeed, the lack of accurate data about the distribution of ground ice affects model development, as well as other research fields. But beyond ice representation, the depth of the LSM also affects subsurface thermodynamics, and in this regard the expansion of the LSMs' depth has improved the simulated permafrost in global climate models (e.g., Nicolsky et al., 2007).*

L382 – As mentioned in previous comment there are borehole temperature measurements in permafrost and at some sites, there are moisture content measurements. There are also often observations of excess ice content when boreholes are drilled.

*Indeed, sometimes you can have some borehole sites with more complete measurements. But the problem is that those extended measurements are seldom available, and that their number is very reduced. Therefore, those sites are very probably not representing global conditions, nor have them a sufficient temporal coverage to include decadal changes in temperature. Such limitations make them, therefore, unsuitable for the scope of our analysis.*

L385 – One of the issues in areas such as the Canadian Arctic is the remoteness and significant cost of drilling boreholes, especially deeper ones where specialized equipment needs to be transported to the site (see for e.g. Smith et al. 2022b). Most permafrost monitoring sites therefore are often located near communities, existing infrastructure, associated with resource development (hydrocarbon, mining) etc.

*Exactly, permafrost monitoring is a complex task because of the difficulty for maintaining the sites and covering such a vast extension of land. We have included this point in the new version of the manuscript.*

L392 – This is also discussed in Smith et al. (2022b) and O'Neill et al. (2020). There are also efforts to

improve ground ice potential modelling and mapping – see for e.g. O'Neill et al. (2019)

*We have included this point in the new version of the manuscript.*

Figure 5 – See previous comments regarding other implications of permafrost thaw such as impacts on infrastructure integrity. Landscape instability is a more inclusive term than ground subsidence.

*We have replaced ground subsidence for landscape instability in Figure 5.*

References cited in comments

Douglas, T. A., Turetsky, M. R. & Koven, C. D. 2020. Increased rainfall stimulates permafrost thaw across a variety of Interior Alaskan boreal ecosystems. npj Clim. Atmos. Sci. 3, 28.

Neumann, R. B. et al. 2019. Warming effects of spring rainfall increase methane emissions from thawing permafrost. Geophys. Res. Lett. 46, 1393–1401.

Noetzli, J. et al. 2022. [Global Climate] Permafrost Thermal State [in "State of the Climate in 2022]; Bull. Am. Met. Soc. Supplement, 103 (8)

O'Neill HB, et al. (2020) Abrupt permafrost thaw and northern development: Comment on "Abrupt changes across the Arctic permafrost region endanger northern development" by B. Teufel and L. Sushama. Nature Climate Change 10:722-723

O'Neill, H. B., Wolfe, S. A. & Duchesne, C. 2019. New ground ice maps for Canada using a paleogeographic modelling approach. Cryosphere 13, 753–773. – See also O'Neill et al.

Phillips, M., et al. (2016). Seasonally intermittent water flow through deep fractures in an Alpine Rock Ridge: Gemsstock, Central Swiss Alps. Cold Regions Science and Technology, 125, 117–127. https://doi.org/10.101

Riseborough D, et al. (2008) Recent advances in permafrost modelling. Permafrost and Periglacial Processes 19 (2):137-156. doi:10.1002/ppp.615

Romanovsky VE, Smith SL, Christiansen HH (2010) Permafrost thermal state in the polar Northern Hemisphere during the International Polar Year 2007-2009: a synthesis. Permafrost and Periglacial Processes 21:106-116

Smith SL, Romanovsky VE, Lewkowicz AG, Burn CR, Allard M, Clow GD, Yoshikawa K, Throop J (2010) Thermal state of permafrost in North America - A contribution to the International Polar Year. Permafrost and Periglacial Processes 21:117-135. doi:10.1002/ppp.690

**References**

Beltrami, H. (2002). Earth's Long-Term Memory. *Science*, **297**(5579), 206–207. DOI: `10.1126/science.1074027`.

Burke, E. J., Zhang, Y., and Krinner, G. (2020). Evaluating permafrost physics in the Coupled Model Intercomparison Project 6 (CMIP6) models and their sensitivity to climate change. *The Cryosphere*, **14**(9), 3155–3174. DOI: `10.5194/tc-14-3155-2020`.

Choulga, M., Kourzeneva, E., Balsamo, G., Boussetta, S., and Wedi, N. (2019). Upgraded global mapping information for earth system modelling: an application to surface water depth at the ECMWF. *Hydrology and Earth System Sciences*, **23**(10), 4051–4076. DOI: `10.5194/hess-23-4051-2019`.

Cuesta-Valero, F. J., García-García, A., Beltrami, H., González-Rouco, J. F., and García-Bustamante, E. (2021). Long-term global ground heat flux and continental heat storage from geothermal data. *Climate of the Past*, **17**(1), 451–468. DOI: `10.5194/cp-17-451-2021`.

García-García, A., Cuesta-Valero, F. J., Beltrami, H., and Smerdon, J. E. (2016). Simulation of air and ground temperatures in PMIP3/CMIP5 last millennium simulations: implications for climate reconstructions from borehole temperature profiles. *Environmental Research Letters*, **11**(4), 044022. DOI: `10.1088/1748-9326/11/4/044022`.

Hermoso de Mendoza, I., Beltrami, H., MacDougall, A. H., and Mareschal, J.-C. (2020). Lower boundary conditions in land surface models – effects on the permafrost and the carbon pools: a case study with CLM4.5. *Geoscientific Model Development*, **13**(3), 1663–1683. DOI: `10.5194/gmd-13-1663-2020`.

Huang, S. (2006). 1851–2004 annual heat budget of the continental landmasses. *Geophysical Research Letters*, **33**(4). n/a–n/a. DOI: `10.1029/2005GL025300`.

Hugelius, G., Strauss, J., Zubrzycki, S., Harden, J. W., Schuur, E. A. G., Ping, C.-L., Schirrmeister, L., Grosse, G., Michaelson, G. J., Koven, C. D., O'Donnell, J. A., Elberling, B., Mishra, U., Camill, P, Yu, Z., Palmtag, J., and Kuhry, P. (2014). Estimated stocks of circumpolar permafrost carbon with quantified uncertainty ranges and identified data gaps. *Biogeosciences*, **11**(23), 6573–6593. DOI: `10.5194/bg-11-6573-2014`.

Koven, C. D., Riley, W. J., and Stern, A. (2013). Analysis of Permafrost Thermal Dynamics and Response to Climate Change in the CMIP5 Earth System Models. *Journal of Climate*, **26**(6), 1877–1900. DOI: `10.1175/JCLI-D-12-00228.1`.

Langer, M., Nitzbon, J., Groenke, B., Assmann, L.-M., Schneider von Deimling, T., Stuenzi, S. M., and Westermann, S. (2022). The evolution of Arctic permafrost over the last three centuries. *EGUsphere*, **2022**, 1–27. DOI: `10.5194/egusphere-2022-473`.

Nicolsky, D. J., Romanovsky, V. E., Alexeev, V. A., and Lawrence, D. M. (2007). Improved modeling of permafrost dynamics in a GCM land-surface scheme. *Geophysical Research Letters*, **34**(8). n/a–n/a. DOI: `10.1029/2007GL029525`.

Obu, J. (2021). How Much of the Earth's Surface is Underlain by Permafrost? *Journal of Geophysical Research: Earth Surface*, **126**(5). e2021JF006123. DOI: https://doi.org/10.1029/2021JF006123.

Pelletier, J. D., Broxton, P. D., Hazenberg, P., Zeng, X., Troch, P. A., Niu, G.-Y., Williams, Z., Brunke, M. A., and Gochis, D. (2016). A gridded global data set of soil, intact regolith, and sedimentary deposit thicknesses for regional and global land surface modeling. *Journal of Advances in Modeling Earth Systems*, **8**(1), 41–65. DOI: https://doi.org/10.1002/2015MS000526.

Pollack, H. N. and Smerdon, J. E. (2004). Borehole climate reconstructions: Spatial structure and hemispheric averages. *Journal of Geophysical Research: Atmospheres*, **109**(D11). n/a–n/a. DOI: 10.1029/2003JD004163.

Slater, A. G. and Lawrence, D. M. (2013). Diagnosing Present and Future Permafrost from Climate Models. *Journal of Climate*, **26**(15), 5608–5623. DOI: 10.1175/JCLI-D-12-00341.1.

Steinert, N., González-Rouco, J., de Vrese, P., García-Bustamante, E., Hagemann, S., Melo-Aguilar, C., Jungclaus, J., and Lorenz, S. (2021). Increasing the Depth of a Land Surface Model. Part II: Temperature Sensitivity to Improved Subsurface Thermodynamics and Associated Permafrost Response. *Journal of Hydrometeorology*, **22**(12), 3231 –3254. DOI: 10.1175/JHM-D-21-0023.1.

Wang, W., Lee, X., Xiao, W., Liu, S., Schultz, N., Wang, Y., Zhang, M., and Zhao, L. (2018). Global lake evaporation accelerated by changes in surface energy allocation in a warmer climate. *Nature Geoscience*, **11**(6), 410–414. DOI: 10.1038/s41561-018-0114-8.

Zhao, G., Li, Y., Zhou, L., and Gao, H. (2022). Evaporative water loss of 1.42 million global lakes. *Nature Communications*, **13**(1), 3686. DOI: 10.1038/s41467-022-31125-6.

---

## Author Comment (AC5)

*Dear Reviewer,*

*We thank you for your thorough and constructive feedback. This file provides a complete documentation of the changes made in response to each of your comments. Reviewer's comments are shown in normal text, author responses are shown in bold, italic, blue text.*

**Reviewer 2**

Cuesta-Valero et al. provide a new estimate of continental heat storage including ground, inland waters and permafrost thawing. For continental heat storage, an update to the previous estimate (Cuesta-Valero et al. 2021) is provided. For inland waters and permafrost thawing, models are used to derive the estimates. I have some major reservations about their methodologies, listed below.

(1). The observation-based estimate for ground heat storage and model-based estimates for inland waters and permafrost thawing are merged together to provide the continental heat storage. I doubt if they can be put together, and then eventually be used in von Schuckmann et al. GCOS assessment (the other components are all observation-based).

*We respectfully disagree with the reviewer on this point. Many relevant studies have combined raw observations, data assimilation (i.e., reanalysis and satellite products), as well as numerical simulations (i.e, global and regional climate simulations) in order to assess the state and evolution of a certain variable of interest. The different Assessment Reports (ARs) of the Intergovernmental Panel on Climate Change (IPCC) are the most important examples of this practice in the climate community. A particular example could be the combination of paleoreconstructions and paleosimulations to obtain a better picture of the last millennium in both IPCC-AR5 (Masson-Delmotte et al., 2013) and IPCC-AR6 (Arias et al., 2021).*

*Furthermore, the rest of components of the GCOS assessment of Earth heat inventory do not include only observational data. For example, the atmosphere heat storage is estimated from reanalysis data, while the heat uptake by glacier melting is estimated from indirect gravimetric observations retrieved from satellites. These two estimates are produced by using numerical techniques and models to assimilate and interpret raw observations. Similarly, we use an observational-based driver to force a numerical model to estimate permafrost heat storage. Concretely, we use ERA-Interim data as upper boundary condition for our permafrost model, which allows us to estimate the ground ice melting that is coherent with the evolution of surface conditions in the last decades. Please note*

*that this is not very different to the use of different reanalyses to estimate the heat storage by the atmosphere in the GCOS paper mentioned by the reviewer.*

*Finally, we must also note that the use of models in our estimates is mostly the result of a lack of adequate data to characterize heat storage in permafrost and inland waters systems. For lakes, for example, existing in-situ data sets with long-term temperature profiles contain only very few lakes relative to the total number of lakes worldwide, and existing repositories are spatially biased towards Europe and North America. For rivers, the data availability is even worse. As we make clear in the manuscript, we incorporate model results because, unfortunately, there are no adequate measurements to derive global estimates of permafrost heat storage and inland waters heat storage.*

*We have also added a new paragraph in the Conclusions stating that although this is not an ideal estimate of continental heat storage, we have identified a clear path toward complementing the use of models with more observational-based estimates.*

(2). Uncertainty estimates for ground heat storage. In this study the uncertainty of the ground heat storage has been reduced by an order compared to their earlier estimate (for example line 200-205). The new estimate suggests a global land heat storage of 84.8 +/- 0.8 mWm-2 (previous estimate is 97+/-6). I found it hard to believe such a small error range, it is simply not possible. Remember you are using only 1000 station data to represent the entire land, even previous error range of 6 is a likely underestimation. I can't understand this small number and I don't understand how this small number is derived given the dataset is basically the same with the previous version.

*Please, note that ground heat storage is estimated from subsurface (borehole) temperature profiles, not from meteorological stations. These temperature-depth profiles record the propagation of alterations in the surface energy balance though the ground, but due to the nature of heat diffusion, borehole profiles are able to retrieve only long-term past changes in surface conditions. That is, decadal to centennial changes in ground heat flux. This reduces greatly the variability in the global average of ground heat storage in our manuscript, even considering the variability at 1079 different locations, thus uncertainty ranges are always going to be narrower than those of estimates based on meteorological stations.*

*Furthermore, we have used a new bootstrap technique to estimate uncertainties from these geothermal data. Previous estimates of global ground heat flux from subsurface temperature profiles provided uncertainty estimates that were biased from a statistical point of view, as they were markedly conservative and included a much wider range than the 95 % confidence interval that is typically*

*provided with the global average. Please, check Cuesta-Valero et al. (2022) for a detailed compari-*
*son of previous uncertainty estimates in comparison with the new bootstrap method, as well as a*
*prove that the uncertainty in previous inversions converge to the one reported here when appropri-*
*ate error propagation methods are considered.*

(3). Uncertainty estimates for permafrost thawing. Only the uncertainty related to the soil thickness
and ice saturation are taken into account. However, I think another major error come from the model
and climate forcing. For example, the use of Mk3L and ERA-Interim, the errors/biases will definitely
propagate into the estimate of this study. I have no idea how to resolve this, as it is related to the
fundamental choices of this study: using models and reanalysis to drive the their estimates.

*Indeed, the use of numerical simulations in the estimates of permafrost heat storage adds uncer-*
*tainties to the results. As discussed in the manuscript, ERA-Interim is now superseded by the ERA5*
*reanalysis, and this new reanalysis should be used in a future iteration of this work. Regarding the*
*Mk3L paleosimulation, new models contributing to the PMIP4 project may be more suitable. In any*
*case, please note that the Mk3L simulation is used only to initialize the permafrost model, as find-*
*ing an equilibrium state for ground ice under preindustrial conditions requires several centuries,*
*and then a transitional period between preindustrial conditions and conditions at the starting date*
*of ERA-Interim (1979 CE) should be provided. Also note that the surface boundary conditions for*
*$\sim 60\%$ of the period of interest are obtained from the observation-based ERA-Interim reanalysis.*
*Therefore, the effect of using a more advanced paleosimulation should be small.*

*The largest uncertainties regarding the permafrost heat uptake are expected to be related to the*
*effect of snow on the ground thermal regime and the distribution of ground ice. These first-order*
*effects were addressed by our parameter ensemble simulations. Please refer to Nitzbon et al. (2022)*
*for an extended discussion of the uncertainties and limitations of the permafrost heat uptake.*

(4). Uncertainty estimates for inland waters. Is the ensemble spread used to estimate the uncertainty of
heat storage in inland waters? If so, it is fundamentally different from the other two components, i.e.
the assumption underlying this method is: model difference (whatever caused the difference) can fully
represent the uncertainty. Such assumption is likely wrong as there are always common model biases.
And such assumption is clearly different from the assumption for your permafrost thawing and ground
heating uncertainty estimate, so they can not be simply added up, simply physically meaningless.

*In our analysis, we explore and analyse each component separately, considering both spatial and*
*temporal variability. Later, a common estimate for the entire continental system is derived. As*
*pointed by the reviewer, using standard error propagation methods to derive the total uncertainty*

*in continental heat storage is excessively simplistic and omits critical differences in the methodology used to derive the heat storage within each subsystem. In the new version of the manuscript, we still provide with an uncertainty estimate for the continental heat storage but explaining the limitations of each method and the differences among the uncertainty estimated for each individual component. We also keep the uncertainty estimates for each individual component; thus the reader can reach their own conclusions about the trustworthiness of the reported uncertainties, and compute its own estimates.*

(5). How the final estimate of land heat storage uncertainty been derived? Are you assuming independency of the three components? Are they independent?

*The final continental heat storage series results from adding the global estimates for ground heat storage, permafrost heat storage, and inland waters heat storage using standard error propagation methods (lines 258-262 in the original manuscript). Here we should inform the reviewer of an error in the processing of the uncertainty estimates that lead to and underestimation of the total uncertainty in continental heat storage. Please, check the new version of the manuscript for an updated estimate.*

*Regarding the independence of the estimates, the three estimates are considered independent, as ground heat storage estimates do not include the heat uptake by permafrost thawing, and no ground heat storage nor permafrost thawing is possible in lakes or reservoirs. Nevertheless, the uncertainty estimate for the total continental heat storage is probably not robust, thus we have added a paragraph discussing the limitations in our estimate (see also our answer to the previous comment).*

(6). Line 219: Please explain why "this large interannual variability is explained by the smaller surface of global lakes and reservoirs in comparison with the global land and permafrost areas"?

*What we wanted to indicate here is that since inland water bodies cover a surface that is two orders of magnitude smaller than the land surface, and one order of magnitude smaller than the total permafrost area, we can expect a larger inter-annual variability in the estimated inland waters heat flux than in the other two components. We have rewritten this sentence in the new version of the manuscript.*

(7). Line 257. The total land heat storage is 23.9+/-0.4 ZJ. The error range is too small to believe. Look at Fig. 1a, there are only several places with observations, and the spatial variability is large (that means you need more data to resolve these variability), so I don't think the uncertainty can be so small. The

uncertainty estimate should be better documented in this study, and any revision should be carefully assessed and validated.

*Please, check our answer to the second comment. Cuesta-Valero et al. (2022) explains in detail the reason for this new smaller uncertainty in global estimates of ground heat flux, and it also includes a comprehensive comparison with previous techniques to retrieve uncertainty from inversions of subsurface temperature profiles. In a nutshell, previous estimates converge to the new uncertainty results when individual inversions from subsurface profiles are aggregated correctly.*

To proceed (avoid rejection of this paper), I recommend the authors not putting the the estimates for the three estimates toghether, just presenting them separately, making a point that permafrost and lakes might be important in EEI, which is the best the authors' can do.. I disagree to put them together because some are model-based estimates, and the uncertainty etsimates are apparant very weak.

*We disagree with the reviewer. We are already providing with the individual estimates for each component of the continental heat storage, analysing their temporal and spatial variability. Nevertheless, we recognize that we underestimated the differences between the sources of uncertainty considered in each continental subsystem, and we have included a discussion about the different uncertainties in each subsystem. Thereby, the readers of the manuscript have access to the individual estimates for ground, permafrost, and inland waters heat storage, information about how to interpret these estimates, the result of applying common error propagation methods, and a warning about the limitations in this uncertainty analysis.*

*Regarding the combination of measurements and models, we refer the reviewer to our answer to the first comment: combination of observation-based results with reanalysis and modelling estimates is a common practice in the climate community when observations for a relevant variable or component of the Earth system are not available. We consider that not using reanalysis or simulations to try to better understand the behaviour of the climate system is a mistake. Nevertheless, we agree that these lines of evidence cannot replace observations, and that they are very different between each other. Therefore, we clearly identify the source of data for our estimates, we clearly indicate in our manuscript that reanalyses and model simulations include additional uncertainties not present in subsurface temperature profiles, and we indicate that observations should be incorporated into the analysis where and when possible. We have also added a paragraph to the Conclusions section clearly indicating that more observation based data should be included in the new version of this analysis.*

*Unfortunately, it is not within our immediate reach to fill the observational gaps appearing in this*

*analysis, but we can still use other sources of information for mitigating those gaps as much as possible.*

**References**

Arias, P., Bellouin, N., Coppola, E., Jones, R., Krinner, G., Marotzke, J., Naik, V., Palmer, M., Plattner, G.-K., Rogelj, J., Rojas, M., Sillmann, J., Storelvmo, T., Thorne, P., Trewin, B., Achuta Rao, K., Adhikary, B., Allan, R., Armour, K., Bala, G., Barimalala, R., Berger, S., Canadell, J., Cassou, C., Cherchi, A., Collins, W., Collins, W., Connors, S., Corti, S., Cruz, F., Dentener, F., Dereczynski, C., Di Luca, A., Diongue Niang, A., Doblas-Reyes, F., Dosio, A., Douville, H., Engelbrecht, F., Eyring, V., Fischer, E., Forster, P., Fox-Kemper, B., Fuglestvedt, J., Fyfe, J., Gillett, N., Goldfarb, L., Gorodetskaya, I., Gutierrez, J., Hamdi, R., Hawkins, E., Hewitt, H., Hope, P., Islam, A., Jones, C., Kaufman, D., Kopp, R., Kosaka, Y., Kossin, J., Krakovska, S., Lee, J.-Y., Li, J., Mauritsen, T., Maycock, T., Meinshausen, M., Min, S.-K., Monteiro, P., Ngo-Duc, T., Otto, F., Pinto, I., Pirani, A., Raghavan, K., Ranasinghe, R., Ruane, A., Ruiz, L., Sallée, J.-B., Samset, B., Sathyendranath, S., Seneviratne, S., Sörensson, A., Szopa, S., Takayabu, I., Tréguier, A.-M., van den Hurk, B., Vautard, R., von Schuckmann, K., Zaehle, S., Zhang, X., and Zickfeld, K. (2021). "Technical Summary". In: *Climate Change 2021: The Physical Science Basis. Contribution of Working Group I to the Sixth Assessment Report of the Intergovernmental Panel on Climate Change*. Ed. by V. Masson-Delmotte, P. Zhai, A. Pirani, S. Connors, C. Péan, S. Berger, N. Caud, Y. Chen, L. Goldfarb, M. Gomis, M. Huang, K. Leitzell, E. Lonnoy, J. Matthews, T. Maycock, T. Waterfield, O. Yelekçi, R. Yu, and B. Zhou. Cambridge, United Kingdom and New York, NY, USA: Cambridge University Press, pp. 33–144. DOI: 10.1017/9781009157896.002.

Cuesta-Valero, F. J., Beltrami, H., Gruber, S., García-García, A., and González-Rouco, J. F. (2022). A new bootstrap technique to quantify uncertainty in estimates of ground surface temperature and ground heat flux histories from geothermal data. *Geoscientific Model Development*, **15**(20), 7913–7932. DOI: 10.5194/gmd-15-7913-2022.

Masson-Delmotte, V., Schulz, M., Abe-Ouchi, A., Beer, J., Ganopolski, A., González Rouco, J., Jansen, E., Lambeck, K., Luterbacher, J., Naish, T., Osborn, T., Otto-Bliesner, B., Quinn, T., Ramesh, R., Rojas, M., Shao, X., and Timmermann, A. (2013). "Information from Paleoclimate Archives". In: *Climate Change 2013: The Physical Science Basis. Contribution of Working Group I to the Fifth Assessment Report of the Intergovernmental Panel on Climate Change*. Ed. by T. Stocker, D. Qin, G.-K. Plattner, M. Tignor, S. Allen, J. Boschung, A. Nauels, Y. Xia, V. Bex, and P. Midgley. Cambridge, United Kingdom and New York, NY, USA: Cambridge University Press. Chap. 5, pp. 383–464. DOI: 10.1017/CBO9781107415324.013.

Nitzbon, J., Krinner, G., von Deimling, T. S., Werner, M., and Langer, M. (2022). *Quantifying the Permafrost Heat Sinkin Earth's Climate System*. Submitted to Geophysical Research Letters. DOI: 10.1002/essoar.10511600.1.

---

## Author Response (AR3)

**Response to reviewers**

Dr. Francisco José Cuesta-Valero on behalf of all coauthors

2023-03-17

*Dear Editor,*

*We thank the anonymous reviewer for the thorough and constructive feedback. This file provides a complete documentation of the changes made in response to each of the reviewer's comments. Reviewer's comments are shown in normal text, including the number associated with the reviewer and the number of the comment. Author's responses are shown in blue, bold, italic text. Indicated line numbers correspond to the version of the manuscript with changes marked unless otherwise indicated.*
* * *
**Reviewer 1**

The detailed response to the review comments is very much appreciated. A number of improvements have been made to the manuscript and for the most part, any comments I had have been adequately addressed. I have suggested a few minor revisions which are either editorial or to improve clarity.

Minor comments:

**Reviewer's Comment 1.1** — L31 – As mentioned in comment on original MS, Obu (2021) provides an equilibrium estimate of permafrost extent (i.e. uses TTOP model). Permafrost that exists due to past climate change including that in the southern portions of the permafrost region are likely underestimated. I still think that you should be clear that this is an equilibrium estimate based on current climate conditions.

**Authors' Response**: *We have modified the text accordingly (line 31).*

**Reviewer's Comment 1.2** — L32-33 – Although it is good that you have added the range considered for these permafrost temperature changes, it only represents a specific decade. You could consider longer-term rates which are updated each year in the state-of-climate reports published in BAMS (most recent Smith et al. 2022), or use the rates reported in Ch 2 of IPCC AR6 WG1 report (Gulev et al. 2021). This would make it clearer to the reader that this warming of permafrost has been going on for several decades and is continuing.

**Authors' Response**: *We have modified the manuscript accordingly (line 34).*

**Reviewer's Comment 1.3** — L-34-37 – Although there have been revisions made to indicate that these estimates only consider "near-surface" permafrost extent, you should also be clear that this refers to 2-3 m depth. A better way to present estimates of permafrost loss would be to give it as a volume as is done in Burke et al. 2020 (and also in the IPCC rept – Fox-Kemper et al. 2021). Estimates are given for volume of permafrost loss per degree of warming which

is probably a more effective way of describing impact of warming on permafrost (less ambiguous and lower chance of misinterpretation than "near-surface permafrost extent").

**Authors' Response**: *The reviewer is right, permafrost volume is less ambiguous than permafrost extent. Nevertheless, permafrost extent is a well-known metric in the community, thus we are now reporting both the changes in permafrost extent and volume (lines 37-38).*

**Reviewer's Comment 1.4** — L37-38 – Be clear here that the latent heat uptake is required for phase change of the ice in permafrost.

**Authors' Response**: *We have modified the text accordingly (line 40).*

**Reviewer's Comment 1.5** — L135 – Check date of Brown et al. The map was published in 1997 as indicated by the citation for the map. This date should probably be given rather than date of access for the digital version. Brown J, Ferrians Jr. OJ, Heginbottom JA, Melnikov ES (1997) Circum-Arctic map of permafrost and ground-ice conditions. U.S. Department of the Interior, U.S. Geological Survey, Map CP-45.

**Authors' Response**: *Done.*

**Reviewer's Comment 1.6** — L131-135 – Does this variation in ice content with depth, take into account that higher ice contents (especially for segregated ice) are usually found near the permafrost table. Some clarification would help.

**Authors' Response**: *For the consideration of excess ice, we relied only on the information provided by the ground ice map of Brown et al. (1997), i.e., we increased the ground ice content in the upper 10-20 m by an amount according to the classification into low/medium/high excess ice content given in the map. As the map by Brown et al. does not provide more detailed information on the vertical distribution of the excess ice, we assumed a uniform distribution of excess ice with depth within that part of the ground. We have also modified the manuscript in order to improve the clarity of this point (lines 136-139).*

**Reviewer's Comment 1.7** — L229-L233 – It might be useful to look at the air temperature data for the region around Hudson Bay and northeastern Canada. My recollection is that there was recent cooling into the 1990s then followed by a period of warming. Spatial variation in temporal change in air temperature may explain some of the patterns regarding heat storage gains. Rouse (1991) might be useful in explaining spatial patterns in heat storage around Hudson Bay.

**Authors' Response**:

*As suggested, we have checked the surface temperature evolution around the Hudson Bay in the CRU TS 4.05, HadCRU 4, and GISSTEMP 4 products (Figure R1). Indeed, all three products display a cooling during the 1990s as indicated by the reviewer (top of the figure). Nevertheless, the magnitude of the cooling is not enough to explain the heat loss in the water bodies of the region, as air temperature trends indicate a long-term warming during the period of interest (i.e., 1960-2020, bottom of the figure).*

*Therefore, this analysis does not explain the negative heat storage in inland water bodies around the Hudson Bay, requiring a more elaborated analysis that could be completed in future iterations of this work.*

[Figure]

**Figure R1:** (top) Surface air temperature anomalies relative to 1960-2020 for the southwestern shore of the Hudson Bay within CRU TS 4.05 data (black), HadCRU 4 data (gray), and GISSTEMP 4 data (red). (bottom) Trends of surface air temperatures during 1960-2020 for the same products.

**Reviewer's Comment 1.8** — L298 – Rather than referring to "abrupt thaw of ground ice" (or ice-rich permafrost), it would be more informative to mention that it is really the changes in the landscape associated with permafrost degradation that exacerbate or enhance the rate of thaw as they do result in a change to the boundary condition by removing material and can also result in increased importance of lateral heat flow (e.g. pond formation and associated collapse of material around pond). Including something on landscape process is important as this really is the "abrupt" part.

**Authors' Response**: *The manuscript has been modified accordingly (line 301).*

**Reviewer's Comment 1.9** — L303 – There can be changes to both water quantity and quality with respect to freshwater conditions. The increased in groundwater flow (and baseflow) will also result in increased mobility of dissolved material that had been immobile in permafrost. This will be important for fish habitat as well as drinking water sources.

**Authors' Response**: *We have specifically mentioned the presence of dissolved materials in frozen water in the new version of the manuscript (line 306).*

**Reviewer's Comment 1.10** — L307-308 – Revision suggested for clarification: "…due to thermokarst processes including ground subsidence, ponding of water, slope instability, riverbank instability and channel widening." (thermokarst processes refer essentially to thawing and settlement of ice-rich permafrost so that the examples given are due to thermokarst processes).

**Authors' Response**: *Thank you for the suggestion, we have incorporated it in the manuscript (lines 312-313).*

**Reviewer's Comment 1.11** — L309 – It is unclear what is meant by "modify traditional construction ways". There were problems in the past because southern construction methods were used that did not consider permafrost. However, since about the mid-twentieth century the potential thaw of permafrost has been considered in engineering design with techniques such as pile foundations, removal of ice-rich material, thick gravel pads as well as use of passive cooling — see for example the large body of work by NRC building research division. Consideration of climate change is more recent, although this has more or less been done for about 25 years, including development of standards and guidelines. It would be better to say that there may be increased infrastructure maintenance costs due to permafrost thaw or something similar.

**Authors' Response**: *We have included the suggested changes in the new version of the manuscript (line 314).*

**Reviewer's Comment 1.12** — L325 – Some clarification is required and it is suggested that "thawing of permafrost" be used instead of referring to "thawing of subsurface ice". Permafrost consists of soil or rock as well as ice. It is largely the organic matter within the soil material that stores the carbon rather than the ice. While I agree as mentioned above, the melting of the excess ice in the permafrost can result in landscape change that can exacerbate thaw, it is more correct to refer to permafrost rather than the ice when referring to carbon.

**Authors' Response**: *We have modified the text accordingly (line 330).*

**Reviewer's Comment 1.13** — L363-364 – Revision suggested "... represented in the Land Surface Model (LSM) components is...." – I think this is what you mean.

**Authors' Response**: *We have incorporated this suggestion in the text (line 367).*

**Reviewer's Comment 1.14** — L367 – revise to "...in LSMs have also improved...." (i.e. LSMs – plural)

**Authors' Response**: *We have fixed this in the new version of the manuscript (line 371).*

**Reviewer's Comment 1.15** — L395-396 – These are borehole temperatures (generally upper 20 to 30 m although some are deeper), rather than surface temperature measurements (although the boreholes are not as deep as the ones used in the determination of ground heat storage in the MS). A revision is therefore required.

**Authors' Response**: *We have fixed this in the new version of the manuscript (line 399).*

**Reviewer's Comment 1.16** — L635 – This paper does not appear to have been accepted and is a "discussion paper" or preprint still undergoing review – maybe indicate it is in review or is a discussion paper?

**Authors' Response**: *We have fixed this reference in the new version of the text, indicating that it is a preprint.*

**Reviewer's Comment 1.17** — Figures 1 and 4 – For clarity revise label to "Permafrost Thaw" so that it is clear that reference is being made to latent heat, rather than the sensible heat considered for "Ground" component in figures.

**Authors' Response**: *We have modified the figures accordingly.*

References

Gulev SK, Thorne PW, et al. (2021) Changing State of the Climate System. In: Masson-Delmotte V, Zhai P, Pirani A et al. (eds) Climate Change 2021: The Physical Science Basis. Contribution of Working Group I to the Sixth Assessment Report of the Intergovernmental Panel on Climate Change. Cambridge University Press, Cambridge, United Kingdom and New York, NY, USA, pp 287-422. doi:10.1017/9781009157896.004

Rouse WR (1991) Impacts of Hudson Bay on the terrestrial climate of the Hudson Bay Lowlands. Arctic and Alpine Research 23:24-30

Smith SL, Romanovsky VE, Isaksen K, Nyland KE, Kholodov AL, Shiklomanov NI, Streletskiy DA, Drozdov DS, Malkova GV, Christiansen HH (2022) [Arctic] Permafrost [in "State of the Climate in 2021"]. Bulletin of the American Meteorological Society 103 (8): S286-S290. doi:10.1175/BAMS-D-22-0082.1

**References**

Brown, J., Ferrians, O., Heginbottom, J., and Melnikov, E. (1997). *Circum-Arctic Map of Permafrost and Ground-Ice Conditions, Version 2*. National Snow and Ice Data Center. National Snow and Ice Data Center [Last accessed: 2023-03-10]. DOI: 10.7265/skbg-kf16.